# Using Confounded Data in Latent Model-Based Reinforcement Learning

**Maxime Gasse**                                                    *maxime.gasse@servicenow.com*
*ServiceNow Research*
*Montréal QC, Canada*

**Damien Grasset**                                        *damien.grasset@irt-saintexupery.com*
*IRT Saint Exupéry Canada*
*Montréal QC, Canada*

**Guillaume Gaudron**                                              *guillaume.gaudron@ubisoft.com*
*Ubisoft La Forge*
*Bordeaux, France*

**Pierre-Yves Oudeyer**                                          *pierre-yves.oudeyer@inria.fr*
*Inria Bordeaux Sud-Ouest*
*Bordeaux, France*

**Reviewed on OpenReview:** *https://openreview.net/forum?id=nFWRuJXPkU*

## Abstract

In the presence of confounding, naively using off-the-shelf offline reinforcement learning (RL) algorithms leads to sub-optimal behaviour. In this work, we propose a safe method to exploit confounded offline data in model-based RL, which improves the sample-efficiency of an interactive agent that collects and learns from online, unconfounded data. First, we import ideas from the well-established framework of *do*-calculus to express model-based RL as a causal inference problem, thus bridging the gap between the fields of RL and causality. Then, we propose a generic method for learning a causal transition model from offline and online data, which captures and corrects the confounding effect using a hidden latent variable. We demonstrate that our method is correct and efficient, in the sense that it attains better generalization guarantees thanks to the confounded offline data (in the asymptotic case), regardless of the confounding effect (the offline expert's behaviour). We showcase our method on a series of synthetic experiments, which demonstrate that a) using confounded offline data naively degrades the sample-efficiency of an RL agent collecting and learning from online data; b) using confounded offline data correctly improves its sample-efficiency.

## 1 Introduction

As human beings, understanding cause and effect is crucial to successfully navigate our environment. If I take aspirin, will my headache go away? Should I better lie down for a while? Should I do both? Two key ingredients in our learning process are observation (we passively contemplate our environment) and experimentation (we perform actions, and we measure their outcomes). While it is well-known that passive observation is not sufficient to infer cause and effect[1], it is hard to believe that it can provide no learning signal at all. The field of cosmology draws models of the universe by exploiting experiments on earth (particle physics) and passive observations of the sky (astronomy). In our everyday lives, the actions of others (e.g., a coworker taking aspirin) can bring some insight into the effects of our own actions. A key question is

---

[1]Simply put, correlation does not imply causation. Or, citing Pearl [29], "behind every causal conclusion there must lie some causal assumption that is not testable in observational studies".

then: which role does observation play when learning cause and effect? This question is at the core of the burgeoning field of causality and reinforcement learning (RL) [2; 40; 16; 41; 42; 43; 26] [2].

In this paper we consider the role of confounded data in the generic setting of model-based RL. Imagine an agent trying to solve a sequential decision-making problem, such as a bot in a videogame. The agent can rely on observational data, collected from the passive observation of other agents (e.g., a dataset of offline traces collected from other bots or humans), and experimental data, collected through the agent's own interactions (e.g., a dataset of online traces collected during learning). We are interested in scenarios where the observational data is confounded, that is, when a hidden variable has been the cause of both actions and their effects in the traces (e.g., when the observed agent had access to privileged information). Our goal is then to understand if and how the use of confounded observational data can improve the sample-efficiency of an online model-based RL agent learning from experimental data.

In the Markov Decision Process (MDP) setting the entire state of the environment is available to the learning agent at each time step, hence there can be no hidden confounder. Because the issue of confounding does not exist, straightforward solutions can leverage large offline datasets safely, leading to the fast-growing field of offline reinforcement learning [20; 21]. In the more general Partially-Observable MDP (POMDP) setting, however, offline data must be considered with more care, because of the potential presence of confounding. A typical example is in the context of medicine, when offline data is collected from physicians who may rely on information absent from their patient's medical records, such as their wealthiness or their lifestyle. Suppose that wealthy patients in general get prescribed specific treatments by their physicians, because they can afford it, while being less at risk to develop severe conditions regardless of their treatment, because they can also afford a healthier lifestyle. This creates a spurious correlation called confounding, and will cause a naive recommender system to wrongly infer that a treatment has positive health effects. Another example is in the context of autonomous driving, when offline data is collected from human drivers who have a wider field of vision than the camera on which the robot driver relies. Suppose human drivers push the brakes when they see a person waiting to cross the street, and only when the person walks in front of the car it enters the camera's field of vision. Then, again, a naive robot might wrongly infer from its observations that whenever the brakes are pushed, a person appears in front of the car. Suppose now that the robot's objective is to avoid collisions with pedestrians, then it might get regrettably reluctant to push the brakes. Of course, in both those situations, the learning agent can infer the right causal model by disregarding the (confounded) offline data altogether, and by relying only on online data instead, collected from its own direct interactions. However, in both those situations also, collecting online data can be expensive, impractical, or even unethical, while collecting offline data by observing the behaviour of human agents is much more affordable.

In this paper we study the question of combining confounded offline data with non-confounded, online data in model-based RL, in the general Partially-Observable Markov Decision Process (POMDP) setting. Our contribution is three-fold:

1. We formalize model-based RL as a causal inference problem using the framework of *do*-calculus [30], which allows us to reason about confounding in the online and offline scenarios in a formal and intuitive manner (Section 3).

2. We present a generic method for combining online and offline data in model-based RL (Section 4), which we demonstrate is correct even when the offline policy relies on privileged hidden information (confounding), and is efficient in the asymptotic case (infinite offline data).

3. We illustrate the effectiveness of our method with a practical implementation for the tabular setting, and three experiments on synthetic toy problems (Section 5).

While our proposed method can be formulated outside of the framework of *do*-calculus, in this paper we hope to demonstrate that it offers a principled and intuitive tool to reason about causality in model-based RL. By relating common concepts from RL and causality, we wish that our contribution will ultimately help to bridge the gap between the two communities.

---

[2]See section 6 for a discussion of related works.

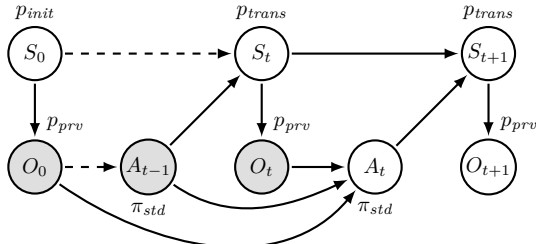
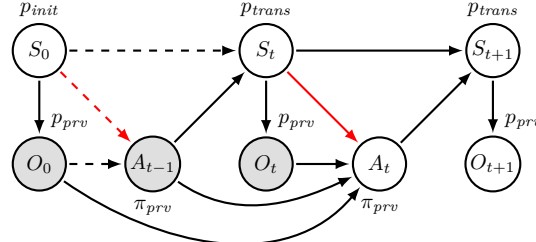

Figure 1: Standard POMDP regime.    Figure 2: Privileged regime (confounding).

## 2 Background

### 2.1 Notation

In this paper, upper-case letters in italics denote random variables (e.g. $X, Y$), while their lower-case counterpart denote their value (e.g. $x, y$) and their calligraphic counterpart their domain (e.g., $x \in \mathcal{X}$). For simplicity we consider only discrete random variables. To keep our notation uncluttered, with a slight abuse of notations we use $p(x)$ to denote sometimes the event probability $p(X = x)$, and sometimes the whole probability distribution of $X$, which should be clear from the context. In sequential models we also distinguish random variables with a temporal index $t$, which might be fixed (e.g., $o_0, o_1$ ), or undefined (e.g., $p(s_{t+1}|s_t, a_t)$ denotes at the same time the distributions $p(s_1|s_0, a_0)$ and $p(s_2|s_1, a_1)$). We also adopt a compact notation for sequences of contiguous variables (e.g., $s_{0 \to T} = (s_0, \dots, s_T) \in \mathcal{S}^{T+1}$ ), and for summations over sets ($\sum_{x \in \mathcal{X}} \iff \sum_x^{\mathcal{X}}$). We assume the reader is familiar with the concepts of conditional independence ($X \perp\!\!\!\perp Y \mid Z$) and probabilistic graphical models based on directed acyclic graphs (DAGs), which can be found in most introductory textbooks, e.g. Pearl [27]; Studeny [36]; Koller and Friedman [18].

### 2.2 Partially-Observable Markov Decision Process

We consider episodic Partially-Observable Markov Decision Processes (POMDPs) of the form $M = (\mathcal{S}, \mathcal{O}, \mathcal{A}, p_{init}, p_{trans}, p_{prv}, r)$, with hidden states $s \in \mathcal{S}$, observations $o \in \mathcal{O}$, actions $a \in \mathcal{A}$, initial and transition state distributions $p_{init}(s_0)$ and $p_{trans}(s_{t+1}|s_t, a_t)$, observation distribution $p_{prv}(o_t|s_t)$, and reward function $r : \mathcal{O} \to \mathbb{R}$. For simplicity we assume episodes $\tau = (o_0, a_0, \dots, o_T)$ of finite length $|\tau| = T > 0$, and we introduce the concept of a history at time $t$, $h_t = (o_0, a_0, \dots, o_t)$. The control mechanism is represented as a stochastic policy $\pi$, which together with the POMDP dynamics $p_{init}$, $p_{trans}$ and $p_{prv}$ defines a probability distribution over trajectories, $p(\tau)$. In this work we consider two types of control policies, which result in two distinct data-generation regimes.

**Definition 1** (Standard POMDP regime)**.** *In the* standard POMDP regime*, actions are decided based only on the visible information from the past, $H_t$, according to a* standard policy $\pi_{std}(a_t|h_t)$*. This results in the data-generation process depicted in figure 1, and trajectory distributions that decompose as*

$$p_{std}(\tau) = \sum_{s_{0 \to |\tau|}}^{\mathcal{S}^{|\tau|+1}} p_{init}(s_0) p_{prv}(o_0|s_0) \prod_{t=0}^{|\tau|-1} \pi_{std}(a_t|h_t) p_{trans}(s_{t+1}|s_t, a_t) p_{prv}(o_{t+1}|s_{t+1}).$$

This standard regime is that of the regular POMDP control problem, which formulates as:

$$\pi_{std}^{\star} = \arg\max_{\pi_{std}} \mathop{\mathbb{E}}_{\tau \sim p_{std}} \left[ \sum_{t=0}^{|\tau|} r(o_t) \right]. \tag{1}$$

**Definition 2** (Privileged POMDP regime)**.** *In the* privileged POMDP regime*, actions can be decided based on the hidden state $S_t$ as well, according to a* privileged policy $\pi_{std}(a_t|h_t, s_t)$*. This results in the data-generation*

*process depicted in figure 2, with trajectory distributions that decompose as*

$$p_{prv}(\tau) = \sum_{s_{0 \to |\tau|}}^{\mathcal{S}^{|\tau|+1}} p_{init}(s_0) p_{prv}(o_0|s_0) \prod_{t=0}^{|\tau|-1} \pi_{prv}(a_t|h_t, s_t) p_{trans}(s_{t+1}|s_t, a_t) p_{prv}(o_{t+1}|s_{t+1}).$$

This privileged regime allows us to consider situations where trajectories are collected by observing an external agent who uses privileged information, in the extreme case the entire POMDP hidden state. Such a privileged agent can be for example a human driver in the context of autonomous driving, who has access to privileged information not accessible to the learning robot, such as the weather forecast. There lies the origin of the confounding problem in offline RL.

### 2.3 Causality and do-calculus

Several frameworks exist in the literature for reasoning about causality [28; 14; 7]. Here we follow the framework of Judea Pearl, whose concept of *ladder of causation* is particularly relevant to answer RL questions. The first level of the ladder, *association*, relates to the passive observation of an external agent acting in the environment, while the second level, *intervention*, relates to the question of what will happen to the environment as a result of the observer's own actions. The tool of *do*-calculus [30], presented in appendix A, acts as a bridge between these two levels, and is typically used to answer whether and interventional distribution, such as $p(y|do(x), z)$, can be identified from an observational distribution, such as $p(x, y, z)$. In a nutshell, in causal systems that can be expressed as DAGs, an intervention $do(x)$ forces the variables in $X$ to take the specific value $X = x$ regardless of their causal ancestors in the graph, and queries of form $p(y|do(x), z)$ measure the effect of an intervention $do(X = x)$ on an outcome event $Y = y$, in the context where another event $Z = z$ is also observed. In this paper, we will use *do*-calculus to reason formally about model-based RL in different POMDP data-collection regimes, which entail different causal graphs.

## 3 Model-based RL as causal inference

Decision-making problems are inherently causal [10; 7]. In POMDPs, model-based RL relies on measuring the causal effect of immediate interventions, $do(a_t)$, on the next observation, $o_{t+1}$, given that past observations, $o_{0 \to t}$, and past interventions, $do(a_{0 \to t-1})$, have already happened. Such causal queries are embodied in the causal transition model $p(o_{t+1}|o_{0 \to t}, do(a_{0 \to t}))$[3], which depends only on the POMDP dynamics in $M$, and not on the control policy $\pi$.

$$p(o_{t+1}|o_{0 \to t}, do(a_{0 \to t})) = p_{std}(o_{t+1}|o_{0 \to t}, do(a_{0 \to t})), \forall \pi_{std}$$
$$= p_{prv}(o_{t+1}|o_{0 \to t}, do(a_{0 \to t})), \forall \pi_{prv}.$$

Together with the initial distribution $p(o_0)$, this causal model allows for the evaluation of any standard control policy $\pi_{std}(a_t|h_t)$. Model-based RL then decomposes the control problem equation (1) into two sub-problems:

1. learning: given a dataset $\mathcal{D}$, estimate a model $\hat{q}(o_{t+1}|h_t, a_t) \approx p(o_{t+1}|o_{0 \to t}, do(a_{0 \to t}))$;

2. planning: given a history $h_t$ and the model $\hat{q}$, derive an optimal action $a_t$.

In this work we consider only the first problem, that is, learning the causal transition model from data. Next, we show using *do*-calculus that this problem can be either trivial or impossible, depending on whether the data is collected using a standard or a privileged control policy.

**Guiding example.** *Consider the door problem illustrated in figure 3. You are sitting in a room with a door, a light that can be red or green, and two buttons that will open the door depending on the light color. You can collect data samples in two ways, either from interventions, i.e., you get up and press the buttons (expensive), or from observations, i.e., you watch someone else press the buttons (cheap). A key detail: you're colorblind*

---

[3]Such a notation can be found also in [26]

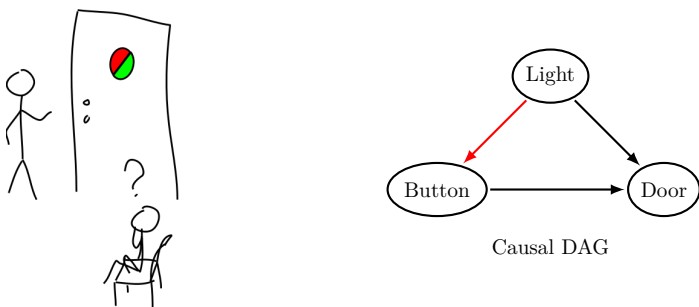

Figure 3: The door problem.

*and can't distinguish red from green. Your goal is to find which button is more likely to open the door. The mechanism responsible for opening the door works as follows: when the light is red, button A opens the door, when the light is green, button B opens the door. The light is red 60% of the time, and green the rest of the time. You are told nothing about the door's mechanism, except that it depends on both the light color and the button pressed (Light → Door ← Button). Since you are colorblind you cannot use the light color to make decisions, and the question you are interested in is simply, which button is more likely to open the door? In the do-calculus framework, this question translates to*

$$\arg\max_{button \in \{A,B\}} p(door{=}open|do(button)).$$

*You have to estimate two causal queries: $p(door{=}open|do(button = A))$ and $p(door{=}open|do(button = B))$.*

### 3.1 In the standard POMDP regime

In the standard POMDP regime, we assume access to a dataset $\mathcal{D}_{std} \sim p_{std}(\tau)$ of episodes $\tau$ collected using an arbitrary standard policy $\pi_{std}(a_t|h_t)$. A key characteristic in this setting is that $A_t \perp\!\!\!\perp S_t \mid H_t$ is always true, that is, every action is independent of the current hidden state given the current history. By applying *do*-calculus on the causal graph from figure 1, the causal model can be shown to be trivially identifiable as

$$p(o_{t+1}|o_{0 \to t}, do(a_{0 \to t})) = p_{std}(o_{t+1}|h_t, a_t). \tag{2}$$

Because of this property, any trajectory $\tau \sim p_{std}(\tau)$ can be interpreted as an *interventional* trajectory, where the learning agent itself could have decided on each of the action $a_t$ in $\tau$. Thus, in the remainder of the paper we will interchangeably call the standard POMDP regime the *interventional regime*, and any dataset $\mathcal{D}_{std}$ collected in this regime an *interventional dataset*.

Assuming sufficient exploration, which is achieved if the control policy is strictly positive ($\pi_{std}(a_t|h_t) > 0$, $\forall a_t, h_t$), an estimator of the POMDP causal model can be obtained from $\mathcal{D}_{std}$ via log-likelihood maximization,

$$\hat{q} = \arg\max_{q \in \mathcal{Q}} \sum_{\tau}^{\mathcal{D}_{std}} \sum_{t=0}^{|\tau|-1} \log q(o_{t+1}|h_t, a_t). \tag{3}$$

This corresponds to the simplest and most common form of model learning via supervised learning [24], which effectively solves our causal inference problem.

**Guiding example.** *Consider again our door example. If you collect the result of your own (or another colorblind person's) interactions with the door, then you know that the light color can not cause which button is pressed (Light $\not\to$ Button). Then, you can directly estimate the causal effect of the button on the door,*

$$p(door{=}open|do(button)) = p_{std}(door{=}open|button).$$

*In this regime, regardless of which policy is used to collect $(button, door)$ samples, eventually you realize that button A has more chances of opening the door (60%) than button B (40%), and thus is the optimal action[4].*

---

[4]One assumption though is strict positivity, $\pi_{std}(button) > 0 \; \forall button$, which ensures that both buttons are pressed.

### 3.2 In the privileged POMDP regime

In the privileged POMDP regime, we assume access to a dataset $\mathcal{D}_{prv} \sim p_{prv}(\tau)$ of episodes $\tau$ collected using an arbitrary privileged policy $\pi_{prv}(a_t|h_t, s_t)$. In this setting, actions might not be independent of the current hidden state given the current history, i.e., $A_t \perp\!\!\!\perp S_t \mid H_t$ might not hold. Because each hidden state $S_t$ has a causal effect on both the current action $A_t$ and the next observation $O_{t+1}$, it acts as a hidden confounder in the POMDP causal transition model. This confounding effect can not be adjusted for without observing the hidden states of the POMDP, and applying *do*-calculus on the causal graph from figure 2 results in the causal model $p(o_{t+1}|o_{0\to t}, do(a_{0\to t}))$ being non-identifiable from $p_{prv}(\tau)$. In particular,

$$p(o_{t+1}|o_{0\to t}, do(a_{0\to t})) \neq p_{prv}(o_{t+1}|h_t, a_t).$$

Because of this, trajectories $\tau \sim p_{prv}(\tau)$ cannot be interpreted as interventional. To better relate to the causality literature, we will interchangeably call the privileged POMDP regime the *observational regime*, and any dataset $\mathcal{D}_{prv}$ collected in this regime an *observational dataset*.

Note that, as a consequence of this non-identifiability, naively applying any off-the-shelf offline RL algorithm [20; 21] on an observational dataset such as $\mathcal{D}_{prv}$ is a risky endeavour, and might result in biased transition models and value functions, and sub-optimal policies.

**Guiding example.** *Take again the door example in figure 3, and assume that you observe someone else interacting with the door. You do not know whether that person is colorblind or not (Light → Button is possible). In this regime, without additional knowledge, you cannot recover the causal queries $p(door=open|do(button))$ from the observed distribution $p(door, button)$. In the do-calculus framework, the queries are said* non identifiable. *However, if that person was to tell you the light color they see before they press A or B, then you could recover those queries via* deconfounding,

$$p(door=open|do(button)) = \sum_{light \in \{red, green\}} p_{prv}(light)p_{prv}(door=open|light, button).$$

*This formula eventually yields the correct causal transition probabilities regardless of the observed policy, given that enough (light, button, door) samples are collected[5].*

### 3.3 Connection to online and offline RL

To relate the concepts of standard (interventional) and privileged (observational) POMDP data to online and offline RL, the key question to ask is, when the samples were collected, could the control policy have used privileged information besides the history $h_t$? Or, more formally, can we guarantee that $A_t \perp\!\!\!\perp S_t \mid H_t$ did hold in the data-generating process?

**In online RL**, the learning agent explicitly controls the data-collection policy, so by design it can not rely on privileged information, hence $A_t \perp\!\!\!\perp S_t \mid H_t$ always holds. Therefore, data collected in an online RL setting can be safely treated as interventional, and the causal transition model can be directly estimated using equation (3).

**In offline RL**, the learning agent might have limited knowledge about the data-collection policy, sometimes no knowledge at all. In some settings, it can be shown that the offline policy could not have used any privileged information, and offline data can be treated as interventional. For example, with human replays from Atari video games, it is hard to imagine a human player having access to more information from the machine's internal state than the regular video and audio outputs from the game. But in more general offline RL settings, access to privileged information can not be dismissed. This is particularly true with human demonstrations collected in the wild, such as in the context of autonomous driving, medical recommender systems (examples in Section 1), or question answering systems [26]. In that case, the offline trajectories can not be considered interventional, and the offline dataset must be treated as observational.

---

[5]The strict positivity condition here is $\pi_{prv}(button|light) > 0 \; \forall button, light$.

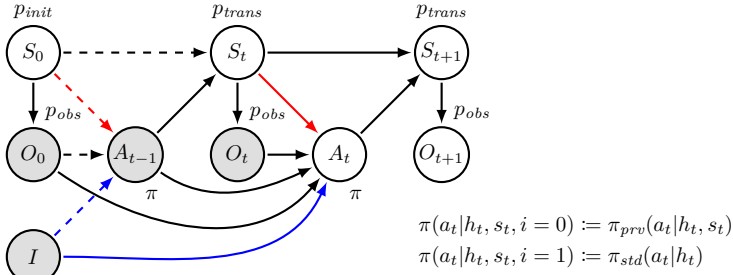

Figure 4: Augmented POMDP setting, with a policy regime indicator $I$ taking values in $\{0, 1\}$ (1=interventional regime, no confounding, 0=observational regime, potential confounding), enforcing the contextual independence $A_t \perp\!\!\!\perp S_t \mid H_t, I = 1$.

## 4 Combining observational and interventional data

Given enough online data, RL agents can learn optimal policies. But in some situations collecting a large online (interventional) dataset can be expensive (recording a robot driver in the wild), while collecting a large offline (observational) dataset from demonstrations is relatively cheap (recording human drivers in the wild). Is it possible then to leverage such offline data to improve the sample-efficiency of an online RL agent, even in the presence of confounding? [6]

### 4.1 Problem statement

We consider two datasets of POMDP trajectories, $\mathcal{D}_{std}$ and $\mathcal{D}_{prv}$, sampled respectively in the standard (interventional) and the privileged (observational) POMDP regime. We then ask the following question: can the observational dataset $\mathcal{D}_{prv}$ be used in combination to the interventional dataset $\mathcal{D}_{std}$, to improve the POMDP causal transition model $p(o_{t+1}|o_{0\rightarrow t}, do(a_{0\rightarrow t}))$ that would be obtained from equation (3) using $\mathcal{D}_{std}$ only? As we will see, answering this question will require to go beyond the identifiability framework of *do*-calculus.

### 4.2 The augmented POMDP

Since both datasets $\mathcal{D}_{std}$ and $\mathcal{D}_{prv}$ are sampled from the same POMDP $(p_{init}, p_{trans}, p_{prv})$ controlled in two different ways, we introduce a regime indicator variable [7] $I \in \{0, 1\}$ that controls an augmented control policy $\pi$. This results in the augmented data-generating process depicted in figure 4, such that

$$\mathcal{D}_{prv} \sim p(\tau|i = 0) := p_{prv}(\tau), \text{ and}$$
$$\mathcal{D}_{std} \sim p(\tau|i = 1) := p_{std}(\tau).$$

Note that the augmented control policy induces the contextual conditional independence $A_t \perp\!\!\!\perp S_t \mid H_t, I = 1$, which is not implied by the DAG factorization. As a direct consequence of equation (2), in this augmented POMDP the causal POMDP transition model can be extracted as

$$p(o_{t+1}|o_{0\rightarrow t}, do(a_{0\rightarrow t})) = p_{std}(o_{t+1}|h_t, a_t) = p(o_{t+1}|h_t, a_t, i = 1). \tag{4}$$

### 4.3 The augmented learning problem

In order to learn the causal transition model $p(o_{t+1}|o_{0\rightarrow t}, do(a_{0\rightarrow t}))$ we propose the following two-step procedure, which relies on fitting a latent probabilistic model $\hat{q}$ that explains both $\mathcal{D}_{std}$ and $\mathcal{D}_{prv}$. Our latent model is constrained to respect the structure of our augmented POMDP, with a latent variable $z_t \in \mathcal{Z}$ that substitutes the true hidden state $s_t \in \mathcal{S}$.

---

[6]Note that we consider this question in its broadest, without further assumptions about the observed offline agent. The offline agent might act sub-optimally, or optimally according to a different reward function than the learning agent.

**Learning.** Our learning problem formulates as standard likelihood maximization[7],

$$\hat{q} = \arg\max_{q \in \mathcal{Q}} \sum_{(\tau)}^{\mathcal{D}_{prv}} \log q(\tau|i=0) + \sum_{(\tau)}^{\mathcal{D}_{std}} \log q(\tau|i=1), \tag{5}$$

with $\mathcal{Q}$ the family of latent probabilistic models that respect the augmented POMDP structure,

$$q(\tau|i=0) = \sum_{z_{0\to|\tau|}}^{\mathcal{Z}^{|\tau|+1}} q_{init}(z_0) q_{obs}(o_0|z_0) \prod_{t=0}^{|\tau|-1} q_{prv}(a_t|h_t, z_t) q_{trans}(z_{t+1}|a_t, z_t) q_{obs}(o_{t+1}|z_{t+1}), \text{ and}$$

$$q(\tau|i=1) = \sum_{z_{0\to|\tau|}}^{\mathcal{Z}^{|\tau|+1}} q_{init}(z_0) q_{obs}(o_0|z_0) \prod_{t=0}^{|\tau|-1} q_{std}(a_t|h_t) q_{trans}(z_{t+1}|a_t, z_t) q_{obs}(o_{t+1}|z_{t+1}).$$

Note that the recovered model $\hat{q}$ conveniently decomposes into a series of simpler components: the initial latent model $\hat{q}(z_0)$, the observation model $\hat{q}(o_t|z_t)$, the latent transition model $\hat{q}(z_{t+1}|z_t, a_t)$, and the behaviour model $\hat{q}(a_t|z_t, h_t, i)$. In practice, the behaviour model in the interventional regime $\hat{q}(a_t|z_t, h_t, i=1) = \hat{q}(a_t|h_t, i=1)$ can be safely ignored during learning, since it does not impact the recovered latent variable nor the causal transition model. Also, the agent behavior model in the observational regime $\hat{q}(a_t|z_t, h_t, i=0)$ can be substituted for a simpler model $\hat{q}(a_t|z_t, i=0)$, which further simplifies the model architecture to be used when solving (5). This leaves only four learnable components: $\hat{q}(z_0)$, $\hat{q}(o_t|z_t)$, $\hat{q}(z_{t+1}|z_t, a_t)$ and $\hat{q}(a_t|z_t, i=0)$, each of which can be approximated using any black-box model, such as a feed-forward neural network.

**Inference.** We recover the causal transition model $\hat{q}(o_{t+1}|o_{0\to t}, do(a_{0\to t})) = \hat{q}(o_{t+1}|h_t, a_t, i=1)$ by applying *do*-calculus on the augmented DAG from figure 4, with $z_t$ instead of $s_t$. The procedure conveniently unrolls as a forward algorithm at test time, and relies on the recurrent computation of $\hat{q}(z_t|h_t, i=1)$, a.k.a. the agent's belief state at time $t$ [5; 35]. First, the initial belief state at $t=0$ is recovered as

$$\hat{q}(z_0|h_0, i=1) = \frac{\hat{q}_{init}(z_0)\hat{q}_{obs}(o_0|z_0)}{\sum_{z_0}^{\mathcal{Z}} \hat{q}_{init}(z_0)\hat{q}_{obs}(o_0|z_0)}.$$

Then, for every $0 \le t < T$, the causal transition model is recovered as

$$\hat{q}(z_{t+1}, o_{t+1}|h_t, a_t, i=1) = \sum_{z_t}^{\mathcal{Z}} \hat{q}(z_t|h_t, i=1)\hat{q}_{trans}(z_{t+1}|z_t, a_t)\hat{q}_{obs}(o_{t+1}|z_{t+1}),$$

$$\hat{q}(o_{t+1}|h_t, a_t, i=1) = \sum_{z_{t+1}}^{\mathcal{Z}} \hat{q}(z_{t+1}, o_{t+1}|h_t, a_t, i=1),$$

and the next belief state is updated to

$$\hat{q}(z_{t+1}|h_{t+1}, i=1) = \frac{\hat{q}(z_{t+1}, o_{t+1}|h_t, a_t, i=1)}{\sum_{z_{t+1}}^{\mathcal{Z}} \hat{q}(z_{t+1}, o_{t+1}|h_t, a_t, i=1)}.$$

Since the observational distribution $p_{prv}$ does not appear in the expression of $p(o_{t+1}|o_{0\to t}, do(a_{0\to t}))$ in equation (4), how does the observational dataset $\mathcal{D}_{prv}$ influence the causal transition model $\hat{q}(o_{t+1}|o_{0\to t}, do(a_{0\to t}))$? The intuition is as follows. The learned model $\hat{q}$ must fit both observational and interventional data by sharing the same latent variables $Z_t$, and the same building blocs $\hat{q}_{init}(z_0)$, $\hat{q}_{obs}(o_t|z_t)$ and $\hat{q}_{trans}(z_{t+1}|z_t, a_t)$. The privileged behaviour model $\hat{q}_{prv}(a_t|z_t)$ is the only component that can allow for discrepancies between the two regimes, and it offers a limited flexibility. As a result, the observational distribution $\hat{q}(\tau|i=0)$ estimated from $\mathcal{D}_{prv}$ acts as a regularizer for the interventional distribution $\hat{q}(\tau|i=1)$ estimated from $\mathcal{D}_{std}$. This regularization helps prevent overfitting when learning from limited interventional data, and improves the generalization performance of the estimated causal transition model. As a side comment, note that our method does not rely on the identifiability of the latent transition model $p_{trans}(s_{t+1}|s_t, a_t)$, which remains in general non-identifiable from observational data, interventional data, or any of their combinations.

---

[7]Note that, while the problem of learning structured latent variable models is known to be hard in general, there also exists a wide range of tools and algorithms available in the literature for solving it approximately, such as the EM algorithm or the method of ELBO maximization.

### 4.4 Theoretical analysis

In this section we analyse the two-step approach described in the previous section, and we demonstrate that it is 1) correct, in the sense that it yields a consistent estimator of the standard POMDP causal transition model and 2) efficient, in the sense that it yields a better estimator than the one based on interventional data only (asymptotically in the number of observational data).

First, let us demonstrate how our approach is correct. An important assumption here is that the latent space of the model is sufficiently large ($|\mathcal{Z}| \geq |\mathcal{S}|$), which ensures enough expressivity to learn the true augmented POMDP distribution, i.e., $p \in \mathcal{Q}$. Under this condition, with enough data $\hat{q}$ converges to $p$, and in particular $\hat{q}(\tau|i=1) \to p(\tau|i=1)$ when $\mathcal{D}_{std} \to \infty$. Then, the standard POMDP causal model $\hat{q}(o_{t+1}|h_t, a_t, i=1)$ being a marginal distribution of $\hat{q}(\tau|i=1)$, it also converges to $p(o_{t+1}|h_t, a_t, i=1)$. Hence, solving equation (5) with a sufficiently large interventional dataset $\mathcal{D}_{std}$ and a sufficiently large latent space $\mathcal{Z}$ converges to the true standard POMDP transition model.

Second, let us demonstrate intuitively how our approach is efficient asymptotically. Our key assumption is that we have a big enough observational dataset, $|\mathcal{D}_{prv}| \to \infty$, which makes it act as a strong regularizer in equation (5). Our key result is theorem 1, which generalizes a famous result in econometrics known as Manski's bounds [23], from the contextual bandit setting ($T = 1$) to the POMDP setting ($T > 1$).

**Theorem 1.** *Assuming $|\mathcal{D}_{prv}| \to \infty$, for any $\mathcal{D}_{std}$ the recovered causal model is bounded as follows:*

$$\prod_{t=0}^{T-1} \hat{q}(o_{t+1}|o_{0\to t}, do(a_{0\to t})) \geq \prod_{t=0}^{T-1} p(a_t|h_t, i=0)p(o_{t+1}|h_t, a_t, i=0), \text{ and}$$

$$\prod_{t=0}^{T-1} \hat{q}(o_{t+1}|o_{0\to t}, do(a_{0\to t})) \leq \prod_{t=0}^{T-1} p(a_t|h_t, i=0)p(o_{t+1}|h_t, a_t, i=0) + 1 - \prod_{t=0}^{T-1} p(a_t|h_t, i=0),$$

*$\forall h_{T-1}, a_{T-1}, T \geq 1$ where $p(h_{T-1}, a_{T-1}, i=0) > 0$.*

*Proof.* See appendix D. $\qquad\square$

Let us denote the family of candidate causal models when solving equation (5) as $\mathcal{H}_0 = \{q(o_{t+1}|o_{0\to t}, a_{0\to t}, i=1) \mid q \in \mathcal{Q}\}$ when $|\mathcal{D}_{prv}| = 0$, and $\mathcal{H}_\infty = \{q(o_{t+1}|o_{0\to t}, a_{0\to t}, i=1) \mid q \in \mathcal{Q} \land q(\tau|i=0) = p(\tau|i=0)\}$ when $|\mathcal{D}_{prv}| \to \infty$. Because there exists at least one episode $\tau = (o_0, a_o, \ldots, o_T)$ with $p(\tau|i=0) > 0$, theorem 1 implies the non-trivial lower bound $q(o_{t+1}|o_{0\to t}, a_{0\to t}, i=1) > 0$ for every $0 \leq t \leq T-1$, at least for the specific values in $\tau$. Therefore, candidate models $q$ such that $q(o_{t+1}|o_{0\to t}, a_{0\to t}, i=1) = 0$ are allowed in $\mathcal{H}_0$ but forbidden in $\mathcal{H}_\infty$, and hence $\mathcal{H}_\infty \subset \mathcal{H}_0$. Because this new hypothesis space is a strict subset of the original one, it offers better generalization bounds for a fixed $|\mathcal{D}_{prv}|$, or equivalently a better sample-efficiency with respect to $|\mathcal{D}_{prv}|$.

**Guiding example.** *Let us now examine our door example in light of Theorem 1. Assume this time that you observe many (button, door) interactions from a non-colorblind person (privileged, $i = 0$), who's policy is $\pi(button=A|light=red) = 0.9$ and $\pi(button=A|light=green) = 0.4$. Then you can already infer from Theorem 1 that $p(door=open|do(button=A)) \in [0.54, 0.84]$ and $p(door=open|do(button=B)) \in [0.24, 0.94]$. You now get a chance to interact with the door ($i = 1$), and you decide to press A 10 times and B 10 times. You are unlucky, and your interventional study results in the following probabilities: $q(door=open|do(button=A)) = 0.5$ and $q(door=open|do(button=B)) = 0.5$. This does not coincide with your (reliable) observational study, and therefore you adjust $q(door=open|do(button=A))$ to its lower bound $0.54$. You now believe that pressing A is more likely to be your optimal strategy.*

### 4.5 Limitations of the provided analysis

We would like to acknowledge two limitations of the theoretical results we provide in the previous section. First, it is fairly easy to see that the upper bound in theorem 1 is not tight. For example,

$$\prod_{t=0}^{T-1} \hat{q}(o_{t+1}|o_{0\to t}, do(a_{0\to t})) \leq \prod_{t=0}^{T-2} \hat{q}(o_{t+1}|o_{0\to t}, do(a_{0\to t})) \quad \forall T \geq 2,$$

is always true, and therefore

$$\prod_{t=0}^{T-1} \hat{q}(o_{t+1}|o_{0\to t}, do(a_{0\to t})) \leq \min_{K \in \{0,\dots,T-1\}} \prod_{t=0}^{K} p(a_t|h_t, i=0) p(o_{t+1}|h_t, a_t, i=0) + 1 - \prod_{t=0}^{K} p(a_t|h_t, i=0)$$

which is a tighter bound and also a generalization of Manski's bounds [23]. Still, it is likely that this upper bound is not tight either. The purpose of theorem 1 is merely to serve as a building block in the argument "observational data creates bounds (in the asymptotic regime) which restrict the hypothesis space for learning". Providing tighter bounds for augmented POMDPs would give valuable insight, and is left for future work.

Second, the results in section 4.4 are restricted to the asymptotic regime $|\mathcal{D}_{prv}| \to \infty$, and do not provide any practical guarantee in the finite-sample regime. Our intuition is that these hard bounds would translate into some kind of soft prior over the hypothesis space, which would also improve generalization. Proving this idea formerly and deriving proper generalization bounds in the finite-sample regime is left for future work.

## 5 Experiments

We run experiments on the three synthetic toy problems described in figure 6, each expressing a different level of complexity and a different form of privileged information. In order to answer the question raised in section 4.1, we compare our *augmented* method against two baselines: *no obs* which discards the observational dataset, and *naive* which naively combines observational and interventional data as if there was no confounding. We expect two things: 1) in the presence of privileged information (confounding), the observational dataset decreases the performance of the *naive* agent, compared to the *no obs* agent; and 2) our *augmented* agent benefits from the observational dataset, and outperforms both the *naive* and the *no obs* agent. The code to reproduce these experiments is available online[8].

### 5.1 Experimental setup

**RL procedure.** Our *augmented* model-based RL procedure is depicted in algorithm 1. We start from a pre-existing dataset of privileged POMDP trajectories (observational data), $\mathcal{D}_{prv}$, an empty dataset of standard POMDP trajectories (interventional data), $\mathcal{D}_{std}$, and a random exploration policy, $\hat{\pi}_{std}$. By fixed increments (e.g., $0, 10, 50, \dots, 1000$), the learning agent collects new interventional trajectories by exploring the environment, which complement the interventional dataset $\mathcal{D}_{std}$. After each increment a new latent-based model $\hat{q}$ is obtained by solving equation (5), and a new (near)-optimal policy $\hat{\pi}_{std}$ is derived from the model $\hat{q}$ using an actor-critic algorithm. The newly obtained policy $\hat{\pi}_{std}$ is then used to collect the next increment of interventional trajectories, supplemented with an $\epsilon$-random noise for exploration.

**Model and agent training.** We train all three model-based methods, *augmented*, *no obs* and *naive*, using the same model architecture and training procedure. Each building bloc $\hat{q}_{init}$, $\hat{q}_{trans}$, $\hat{q}_{obs}$ and $\hat{q}_{prv}$ consists in a tabular logistic model, and equation (5) is solved via mini-batch stochastic gradient descent using Adam [17]. Once the POMDP dynamics are recovered we extract $\hat{q}(o_0)$ and $\hat{q}(o_{t+1}|o_{0\to t}, do(a_{0\to t}))$ to train a "dreamer" agent [11] via actor-critic, implemented as a feed-forward neural network that takes as input the recovered POMDP belief state, $\hat{q}(z_t|o_{0\to t}, do(a_{0\to t-1}))$.

**Evaluation.** We evaluate on the real test environment 1) the quality of the causal transition model $\hat{q}(o_{t+1}|o_{0\to t}, do(a_{0\to t}))$ in terms of its likelihood on new interventional data (collected via random exploration),

---

[8]Code for the experiments: `https://github.com/gasse/causal-rl-tmlr`

---

**Algorithm 1** Augmented model-based RL pseudocode

---

**Require:** observational dataset $\mathcal{D}_{prv}$ (potentially privileged), training method (`augmented`, `no_obs` or `naive`), training steps $N$ (e.g., $N = (0, 10, 50, \dots, 1000)$), exploration noise $\epsilon$
**Ensure:** estimated POMDP dynamics $\hat{q}_{init}(z_0)$, $\hat{q}_{obs}(o_t|z_t)$, $\hat{q}_{trans}(z_{t+1}|z_t, a_t)$, and optimal policy $\hat{\pi}_{std}$
    Initialize $\mathcal{D}_{std} \leftarrow \emptyset$, $\hat{\pi}_{std} \leftarrow$ random exploration policy
    **if** training method is `no_obs` **then** discard observations ($\mathcal{D}_{prv} \leftarrow \emptyset$)

    **if** training method is `naive` **then** consider observations as interventions ($\mathcal{D}_{std} \leftarrow \mathcal{D}_{prv}$ and $\mathcal{D}_{prv} \leftarrow \emptyset$)

    $n_{prev} \leftarrow 0$
    **for all** $n \in N$ **do**
        Collect $n - n_{prev}$ new interventional samples using $\hat{\pi}_{std} + \epsilon$-random noise, and add them to $\mathcal{D}_{std}$
        Obtain a new model $\hat{q}$ using $\mathcal{D}_{std}$ and $\mathcal{D}_{prv}$ (equation (5), supervised learning)
        Obtain a new policy $\hat{\pi}_{std}$ using the model $\hat{q}$ (actor-critic in "dream" environment)
        $n_{prev} \leftarrow n$

---

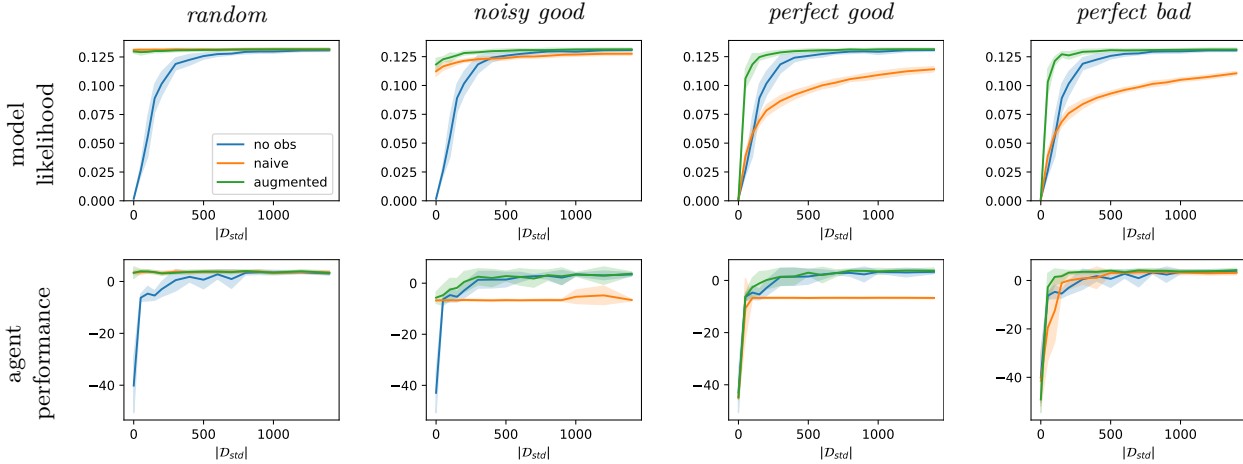

Figure 5: Robustness to different degrees of confounding on the `tiger` problem. **Top**: model likelihood on new data (higher is better). **Bottom**: agent performance in terms of cumulated reward (higher is better). **Columns**: the different privileged agents used to collect the observational dataset $\mathcal{D}_{obs}$. The *random* agent amounts to no confounding, while the *noisy good*, *perfect good* and *perfect bad* agents introduce different degrees of confounding. We report the mean $\pm$ standard deviation over 10 random seeds. Our *augmented* method performs best regardless of the degree of confounding, both in terms of model likelihood and agent performance, and is on par with the *naive* method when there is no confounding (random).

and 2) the performance of the resulting policy $\hat{\pi}_{std}$ in terms of its cumulated reward. We evaluate each model and agent over 10K new trajectories, and we repeat each experiment 10 times with different random seeds to account for variability. We defer the reader to appendix B for the complete experimental details.

## 5.2 Robustness to different degrees of confounding in the tiger problem

`Tiger` is a classic small-scale POMDP from Cassandra et al. [4] with $|\mathcal{S}| = 6$ hidden states and time horizon $T = 10$. To measure the robustness of our method to different degrees of confounding, we consider different privileged policies, described in detail in appendix B. For each experiment we collect $|\mathcal{D}_{prv}| = 1500$ (confounded) observational trajectories, and we train and evaluate at increments $|\mathcal{D}_{std}| \in (0, 50, 100, 150, 200, 300, 400, 500, 600, 700, 800, 900, 1000)$.

Results are reported in figure 5. The first policy, *random*, amounts to a degree 0 of confounding, because the privileged information is effectively not used to generate $\mathcal{D}_{prv}$. In this setting our method performs on par with the *naive* method, which also leverages the observational data without bias due to the absence of

confounding. The *noisy good* policy mostly opens the correct door with the treasure, but sometimes also decides to listen or to open the wrong door at random. This degree of confounding, although rather mild in terms of the bias in the non-causal transition probabilities $p_{prv}(o_{t+1}|h_t, a_t)$, is particularly hurtful to the *naive* method. The *perfect good* policy always opens the correct door, which induces non-causal transition probabilities that are completely off the causal ones, and also hurts the performance of the *naive* method as the learned model will tend to be over-optimistic. The *perfect bad* policy always opens the wrong door, which induces transition probabilities that are completely off in the other direction, but this time the effect on the performance of the *naive* method is not as bad. Indeed, the learned *naive* model this time will tend to be over-pessimistic, which is not such a bad prior in the case of the `tiger` problem. Our *augmented* method, always performs better than (or on par with) both *no obs* and *naive*, effectively leveraging the observational dataset $|\mathcal{D}_{prv}|$ even in the presence of confounding.

### 5.3 Performance on the gridworld problems

`Hidden treasures` is a 3x3 grid-world problem inspired from Sutton et al. [37], with $|\mathcal{S}| = 36$ hidden states and a time horizon $T = 10$. `Sloppy dark room` is a 5x5 grid-world inspired from Alt et al. [1], with $|\mathcal{S}| = 21$ hidden states and a time horizon $T = 30$. In both problems, the privileged agent has complete information at each time step, and uses a shortest path algorithm to decide on its next action. For each experiment we collect $|\mathcal{D}_{prv}| = 8000$ (confounded) observational trajectories, and we train and evaluate each method at increments $|\mathcal{D}_{std}| \in (25, 50, 75, 100, 150, 200, 300, 400, 600, 800, 1000, 1500, 2000, 3000, 4000, 5000, 6000)$.

Results are reported in figure 7, where a similar trend can be observed in the two experiments. Initially, when few interventional samples have been collected, the observational data seems to benefit the *naive* method, despite the presence of confounding, and it exhibits gains both in terms of model likelihood and agent performance compared to the *no obs* method. But as more samples are collected, the untreated confounding effect in the observational data starts becoming hurtful to the *naive* method, and eventually better performances are obtained by disregarding the observational data, using the *no obs* method. In both cases our *augmented* method makes the best use of the available data, and exhibits a better convergence rate than both *naive* and *no obs*, in terms of model quality and agent performance, despite the confounding effect present in the observational data. In figure 7, right side, we report the density of tiles visited by the agent of each method, at a chosen time step where our *augmented* method has almost converged, while the two other methods haven't. In `hidden treasures` we see that our method has learned to cycle through all 4 corners, which is the optimal strategy, while the two other methods still struggle and focus only on a single side or a single corner of the grid. In `sloppy dark room`, our method succeeds in quickly escaping the upper section and reaching the target, while the two other methods struggle more in the upper section, and reach the target less often. It is interesting to observe a very similar outcome in both experiments, while the type of privileged information, hidden to the learning agent, is quite different. In `hidden treasures` the learning agent knows its position at all times, but is missing information about where the target is located, thus it needs to explore the space at test time, while the privileged agent just has to go straight to the target. In `sloppy dark room`, the target is fixed hence no exploration of the space is needed at test time, but the agent's location is hidden most of the time and the agent has to learn how to best navigate while being half-blind.

## 6 Related work

**Causal RL.** A whole body of work exists around the question of merging interventional and observational data in RL in the presence of confounding. Bareinboim et al. [2] study a sequential decision problem similar to ours, but assume that expert intentions are observed both in the interventional and the observational regimes, i.e., prior to doing interventions the learning agent can ask "what would the expert do in my situation?" This artificially introduces an intermediate, observed variable $\hat{a}_t = f(o_t)$ with the property that $p_{prv}(a_t = \hat{a}_t|\hat{a}_t) = 1$, which effectively removes any confounding ($A_t \perp\!\!\!\perp S_t|H_t$). Zhang and Bareinboim [40; 43] relax this assumption in the context of binary bandits, and later on in the more general context of dynamic treatment regimes [41; 42]. They derive causal bounds similar to ours (Theorem 1), and propose a two-step approach which first extracts causal bounds from observational data, and then uses these bounds in an online RL algorithm. While their method nicely tackles the question of leveraging observational data

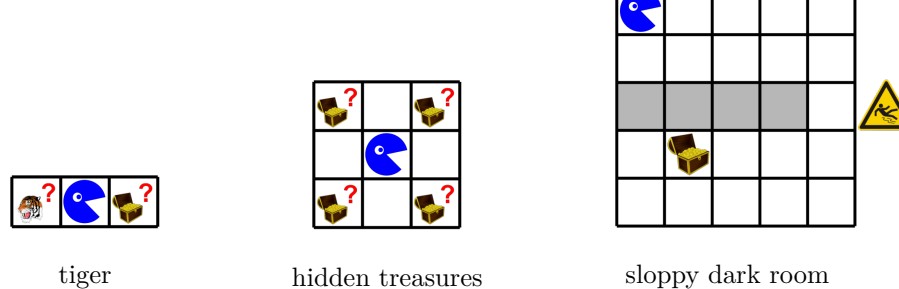

tiger          hidden treasures          sloppy dark room

Figure 6: Our three synthetic toy problems. In **tiger**, the learning agent receives a noisy signal of the tiger's position (roar left or roar right). It can wait and listen to a new roar at the cost of -1 reward, or decide to move left or right. Treasure gives +10 reward, and tiger gives -100 reward. The privileged agent knows the exact location of the tiger. The game stops when treasure or tiger is found, or after a maximum horizon of $T = 10$. In **hidden treasures** the agent must collect a treasure (+1 reward), which is randomly located in one of the four corners. The privileged agent knows the treasure's position at all times, the learning agent doesn't. The treasure is reset to a new location when found, and the game stops after a fixed horizon of $T = 10$. In **sloppy dark room** the agent must reach a treasure (+1 reward) located behind a wall, and slips to a random adjacent tile instead of moving to the chosen direction 50% of the time. The privileged agent knows its position at all times, while the learning agent is only revealed its position with 20% chances at each time step, and is blind otherwise (a dummy position is revealed). The time horizon is fixed to $T = 30$.

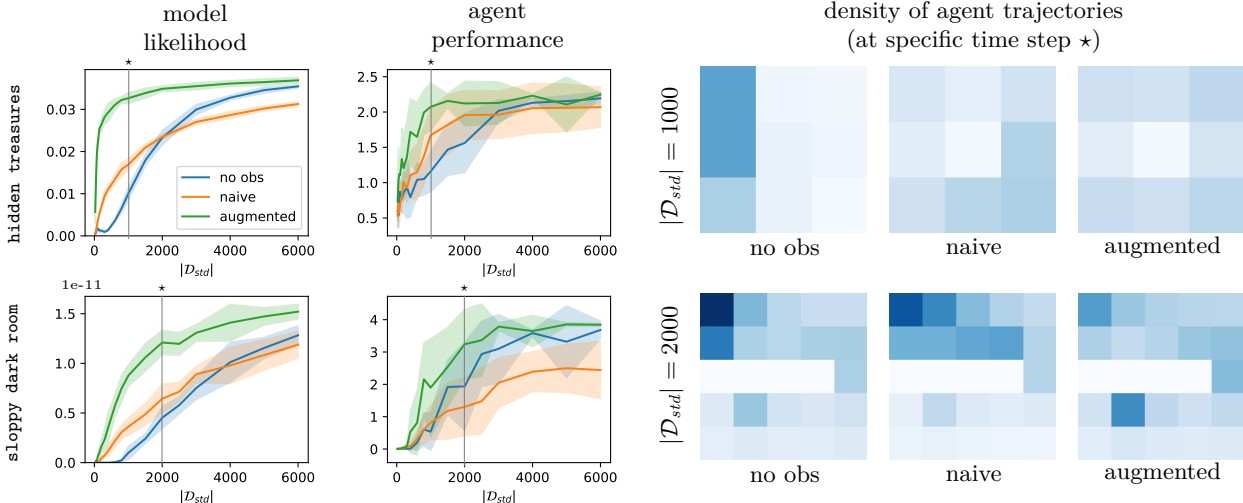

Figure 7: Performance on the `hidden treasures` (**top row**) and `sloppy dark room` (**bottom row**) grid-world problems. **First column**: model likelihood on new data (higher is better). **Second column**: agent performance in terms of cumulated reward (higher is better). **Third column**: density of the agent trajectories obtained by each method at a specific time step $\star$. In both setups, our *augmented* method outperforms both the unbiased *no obs* method and the biased *naive* method, and displays a better sample-efficiency in terms of interventional (online) data due to its correct use of the observational (offline, confounded) data. While the *naive* method can sometimes provide some initial gains compared to the *no obs* method, at some point the confounding effect in the observational dataset (collected from a privileged agent) starts affecting the agent's performance negatively, and one is better off not using the confounded data at all (*no obs* method).

for online exploration, it does not account for uncertainty in the bounds estimated from the observational data. In comparison, our latent-based approach is more flexible, as it never computes explicit bounds, but rather lets the learning agent balance through equation (5) how data from both regimes will influence the final transition model, depending of the number of samples available. Kallus et al. [16] also propose a two-step procedure to combine observational and interventional data, which however requires a series of strong parametric assumptions (strong one-way overlap, linearity, non-singularity etc.), and only works in the context of binary contextual bandits.

**Causal confusion.** In the context of imitation learning, the problem of *causal misidentification*, that is, ascribing the actions of the expert to the wrong explanatory variables, is attributed to confounding by de Haan et al. [8]. Spencer et al. [34] argue instead that it is a manifestation of *covariate shift*, which appears more plausible to us. Indeed, it can be shown that in the experiments of de Haan et al. [8] the experts don't use privileged information, which theoretically circumvents any confounding.

**Exploiting offline RL data.** Combining online and offline data in RL also raises additional challenges, such as the value function initialisation problem [9] or the bootstrapping error accumulation problem [19; 25]. While these challenges could be combined with and amplified by confounding, they originate from fundamentally different issues and require orthogonal treatments. Off-policy evaluation, which is about estimating the performance of a policy $\pi$ using observational data only, can be seen as a specific instantiation of the framework we present in this paper. It corresponds to the particular setting $|\mathcal{D}_{int}| = 0$, where it can be shown that the causal transition model is in general not recoverable in the presence of confounding. Still, a growing body of literature studies the question of learning purely from offline data in the presence of confounding, under the assumption that the data-generating process respects specific structural or parametric constraints [22; 38; 3].

**Large sequence models.** A recent trend in RL is to apply large sequence models to estimate the environment's dynamics in a model-based fashion [32; 15], or to parameterize a goal-conditioned policy in a model-free fashion [6; 44]. While large sequence models appear a promising tool for efficiently combining offline and online datasets, they remain vulnerable to confounding, as pinpointed by Ortega et al. [26]. Because our proposed method follows a generic model-based approach, in theory it could be easily combined with a large sequence model to address large-scale RL scenarios, while remaining robust to confounding.

## 7 Discussions

In this paper we have presented a simple, generic method for combining interventional and observational (potentially confounded) POMDP trajectories in model-based reinforcement learning. We have demonstrated theoretically that our method is correct and efficient in the asymptotic case (infinite observational data), and we have illustrated its effectiveness empirically on three synthetic toy problems. We have also highlighted the dangers of naively using offline data collected under privileged information in RL, which can effectively hurt the performance of an online RL agent, and reduce its sample-efficiency. The main limitation of our method is that it adds an additional challenge on top of model-based RL, that of learning a latent-based POMDP transition model, which can become problematic in high-dimensional RL settings. Still, the recent success of discrete latent-based models for solving complex RL tasks [11; 39; 12] or tasks in high-dimensional domains [31] lets us envision that this difficulty can be overcome in practice. A first extension of this work would be to develop a similar approach using latent-free transition models, which would remove the challenge of learning a latent variable model. This seems doable at least in the case $T = 1$. A second potential extension to this work could be to consider a setting where several privileged agents are observed, each with its own distinct policy, leading to multiple observational datasets. This would lead, in the asymptotic case, to a stronger regularizer for the causal POMDP transition model, as the implicit bounds implied by theorem 1 would combine into tighter ones. A third, obvious extension would be to develop a similar approach within the framework of model-free RL, which could take the form of an explicit or implicit value-function regularizer. A fourth direction to investigate is that of scaling, by extending our method to RL tasks with continuous observation spaces (e.g., pixel-based) and continuous latent spaces, so that it can be applied to a broader range of problems. Finally, we hope that this work will help bridge the gap between the fields of RL and causality, and will convince the RL community that causality is an adequate tool to reason about offline, observational data, which is plentiful in the world.

## 8    Acknowledgements

We thank the anonymous TMLR reviewers for their comments which helped us improve the quality of this work. We thank David Berger for early discussions and for pointing us to relevant bodies of work. This work was supported by the Canada First Research Excellence Fund (CFREF), Canada Excellence Research Chairs (CERC), and the DEpendable Explainable Learning (DEEL) transatlantic research program.

## Broader Impact Statement

Confounding is a prevalent issue in human-generated data, and can be an important source of bias in the design of decision policies, if not dealt with properly. This paper makes a humble step towards more robustness and fairness in AI-based decision systems, by combining causality and statistical learning to address the confounding problem in reinforcement learning from offline data. As such this work has potentially important societal implications, in particular in critical systems where lives are at stake, such as medicine or self-driving cars, and where human-generated data is prevalent.

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

# Appendices

## A    Introduction to do-calculus

The framework of *do*-calculus [30] was proposed as an intuitive tool to answer identifiability questions given a causal graph $\mathcal{G}$, such as, can the interventional distribution $p(y|do(x), z)$ be recovered from the observational distributions $p(y, x, z)$?

### A.1    The three rules of do-calculus

Do-calculus relies on three graphical rules, which depend solely on the existence of specific structural constraints in $G$:

- R1: insertion/deletion of observations, $p(y|do(x), z, w) = p(y|do(x), w)$ if $Y$ and $Z$ are *d*-separated by $X \cup W$ in $\mathcal{G}^\star$, the graph obtained from $\mathcal{G}$ by removing all arrows pointing into variables in $X$.

- R2: action/observation exchange, $p(y|do(x), do(z), w) = p(y|do(x), z, w)$ if $Y$ and $Z$ are *d*-separated by $X \cup W$ in $\mathcal{G}^\dagger$, the graph obtained from $\mathcal{G}$ by removing all arrows pointing into variables in $X$ and all arrows pointing out of variables in $Z$.

- R3: insertion/deletion of actions, $p(y|do(x), do(z), w) = p(y|do(x), w)$ if $Y$ and $Z$ are *d*-separated by $X \cup W$ in $\mathcal{G}^\ddagger$, the graph obtained from $\mathcal{G}$ by first removing all the arrows pointing into variables in $X$ (thus creating $\mathcal{G}^\star$) and then removing all of the arrows pointing into variables in $Z$ that are not ancestors of any variable in $W$ in $\mathcal{G}^\star$.

This set of rules has been shown to be complete [13; 33], and results in an algorithm polynomial in the number of nodes in $\mathcal{G}$ to answer identifiability questions, which either outputs "no" or "yes" along with an estimate (a recovery formula) based on observational quantities. We refer the reader to Pearl [30] for a thorough introduction to *do*-calculus.

### A.2    Note on ignorability and exogeneity

In this paper we use at great length the concept of confounding, which is a core idea in Judea Pearl's *do*-calculus framework. For readers who are more familiar with the framework of potential outcomes from Donald Rubin [14], the concept of confounding closely relates to the concepts of ignorability and exogeneity, which can be shown to be equivalent to the unconfoundedness (no confounding) assumption [28].

## B   Experimental details

### B.1   Training

In all our experiments we use tabular logistic models for each of the components in $\hat{q}$. That is, each building bloc $q(z_0)$, $q(o_t|z_t)$, $q(z_{t+1}|z_t, a_t)$, and $q(a_t|h_t, z_t, i = 0)$ is parameterized using a set of softmax-normalized scalars vectors. We train $\hat{q}$ via gradient descent using the Adam optimizer [17], by directly minimizing the negative log likelihood of the model (equation (5)) on random mini-batches of trajectories sampled from $\mathcal{D}_{std} \cup \mathcal{D}_{prv}$. Agents are trained using the learned model as a "dream" environment (by sampling imaginary trajectories $\tau \sim \hat{q}(\tau|i = 1)$), with a simple actor-critic algorithm (REINFORCE with a state-value baseline) for a fixed number of iterations, also using the Adam optimizer. Both the actor and critic consists of a 2-layers perceptron (MLP) with the same hidden layer size, which take as input the belief state recovered from the model. The training hyperparameters we used in each experiment are displayed in table 1.

|  | tiger | hidden treasures | sloppy dark room |
|---|---|---|---|
| **Latent model** | | | |
| latent space size $|\mathcal{Z}|$ | 32 | 256 | 128 |
| learning rate | $10^{-3}$ | $10^{-3}$ | $10^{-3}$ |
| number of epochs (max) | 500 | 500 | 500 |
| number of gradient steps per epoch | 50 | 100 | 100 |
| minibatch size (trajectories $\tau$) | 32 | 64 | 64 |
| **Actor-critic agent** | | | |
| exploration noise $\epsilon$ | 0.5 | 0.2 | 0.2 |
| hidden layer size | 256 | 512 | 256 |
| learning rate | $5 \times 10^{-4}$ | $5 \times 10^{-4}$ | $5 \times 10^{-4}$ |
| number of epochs | 200 | 400 | 200 |
| number of gradient steps per epoch | 50 | 50 | 50 |
| minibatch size (trajectories $\tau$) | 32 | 64 | 64 |
| minibatch return scaling | yes | no | no |
| entropy bonus | $10^{-3}$ | $10^{-3}$ | $10^{-3}$ |
| discount factor $\gamma$ | 1 | 1 | 1 |

Table 1: Training hyperparameters we used in each experiment. When learning the model, we divide the learning rate by 10 after 10 epochs without loss improvement (reduce on plateau), and we stop training after 20 epochs without improvement (early stopping). We use all available data for training, and we monitor the training loss for early stopping (no validation set).

### B.2   Evaluation

**Model quality (likelihood).**   To evaluate the general quality of the recovered POMDP model, we compute the likelihood of $\hat{q}$ on a new interventional dataset $\mathcal{D}_{test}$ obtained from the true environment $p$ with a uniformly random policy $\pi_{rand}$,

$$\mathbb{E}_{\tau \sim p_{init}, p_{trans}, p_{obs}, \pi_{rand}} \left[ \hat{q}(o_0) \prod_{t=1}^{|\tau|} \hat{q}(o_{t+1}|h_t, i = 1) \right].$$

We report an empirical estimate of this measure using 10000 trajectories.

**Agent performance (cumulated reward).**   To evaluate quality of the agent obtained from the model $\hat{q}$ for solving the standard POMDP control task, we compute the expected cumulated reward of the policy $\hat{\pi}^{\star}$

on the true environment $p$,

$$\mathbb{E}_{\tau \sim p_{init}, p_{trans}, p_{prv}, \hat{\pi}^{\star}} \left[ \sum_{t=1}^{|\tau|} r(o_t) \right].$$

We report an empirical estimate of this measure using 10000 trajectories.

### B.3 Tiger experiment

We present the (compact) POMDP dynamics of the `tiger` problem in table 2. After conversion to the notation in the paper, the observations become $o_t = (roar_t, reward_t)$, the actions remain $a_t = action_t$, and the hidden states are $s_t = (tiger_t, reward_t)$. The privileged policies used in the experiments (section 5.2) are reported in table 3.

Table 2: Compact POMDP dynamics in the `tiger` problem.

| $tiger_0$ | |
|---|---|
| left | right |
| 0.5 | 0.5 |

$p(tiger_0)$

| $tiger_t$ | $roar_t$ | |
|---|---|---|
| | left | right |
| left | 0.85 | 0.15 |
| right | 0.15 | 0.85 |

$p(roar_t|tiger_t)$

| $tiger_t$ | $action_t$ | $tiger_{t+1}$ | |
|---|---|---|---|
| | | left | right |
| left | listen | 1.0 | 0.0 |
| | open left | 0.5 | 0.5 |
| | open right | 0.5 | 0.5 |
| right | listen | 0.0 | 1.0 |
| | open left | 0.5 | 0.5 |
| | open right | 0.5 | 0.5 |

$p(tiger_{t+1}|tiger_t, action_t)$

| $tiger_t$ | $action_t$ | $reward_{t+1}$ | | |
|---|---|---|---|---|
| | | -1 | -100 | +10 |
| left | listen | 1.0 | 0.0 | 0.0 |
| | open left | 0.0 | 1.0 | 0.0 |
| | open right | 0.0 | 0.0 | 1.0 |
| right | listen | 1.0 | 0.0 | 0.0 |
| | open left | 0.0 | 0.0 | 1.0 |
| | open right | 0.0 | 1.0 | 0.0 |

$p(reward_{t+1}|tiger_t, action_t)$

Table 3: Privileged policies $\pi_{prv}(action|tiger)$ used in the `tiger` experiment.

| privileged policy | $tiger_t$ | $action_t$ | | |
|---|---|---|---|---|
| | | listen | left | right |
| random | left | 0.33 | 0.33 | 0.33 |
| | right | 0.33 | 0.33 | 0.33 |
| noisy good | left | 0.05 | 0.30 | 0.65 |
| | right | 0.05 | 0.80 | 0.15 |
| perfect good | left | 0.00 | 0.00 | 1.00 |
| | right | 0.00 | 1.00 | 0.00 |
| perfect bad | left | 0.00 | 1.00 | 0.00 |
| | right | 0.00 | 0.00 | 1.00 |

## C  Additional empirical results

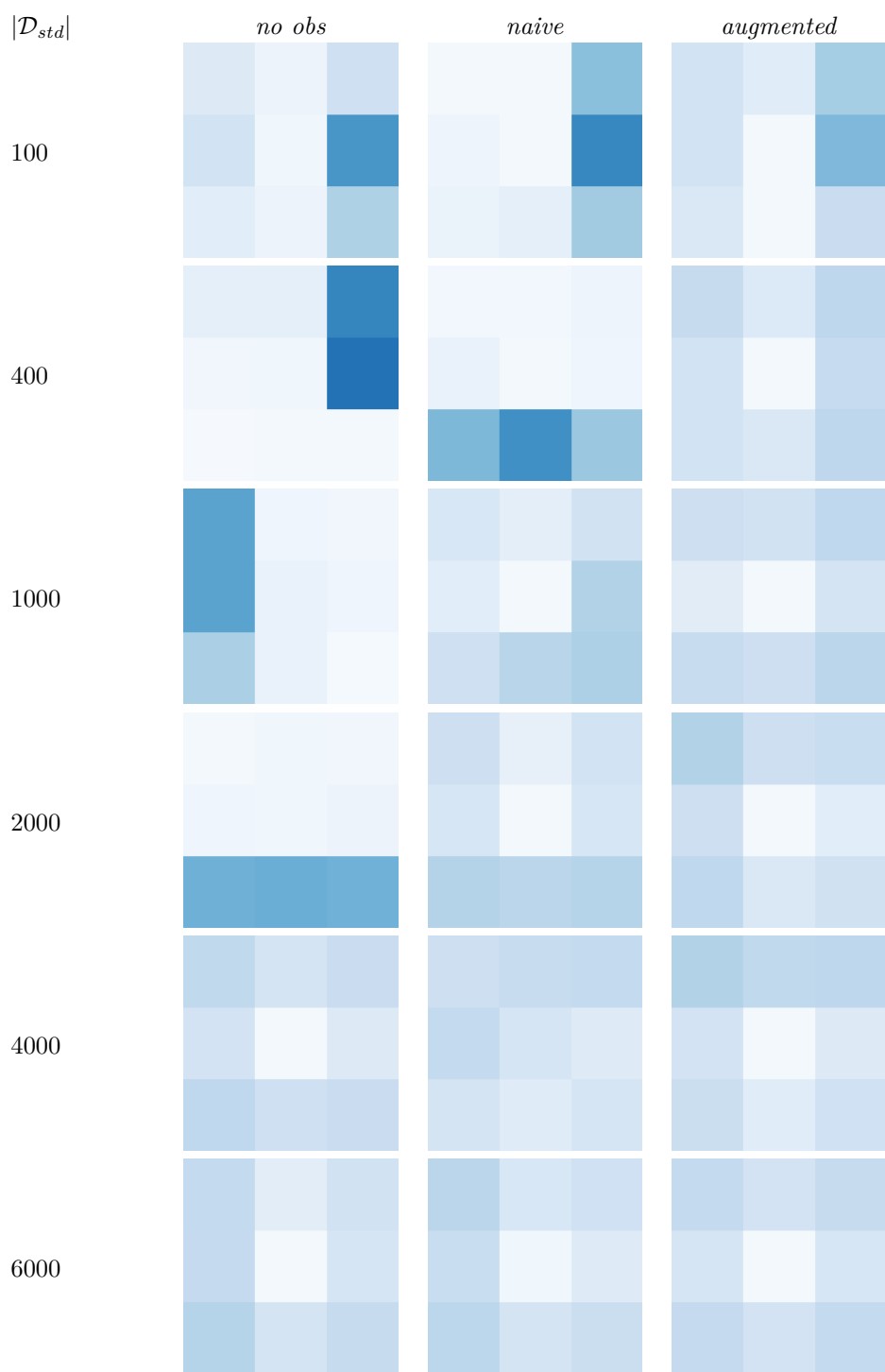

Figure 8: Evolution of the test-time agent trajectories in the `hidden treasures` experiment. We report a heatmap of the tiles visited by each agent (*no obs*, *naive*, *augmented*) at different time steps (number of interventional samples collected) during a single RL run (single seed). Eventually all methods converge to the optimal strategy, which is to cycle through the 4 corners. Our *augmented* method converges to this behaviour earlier on during training.

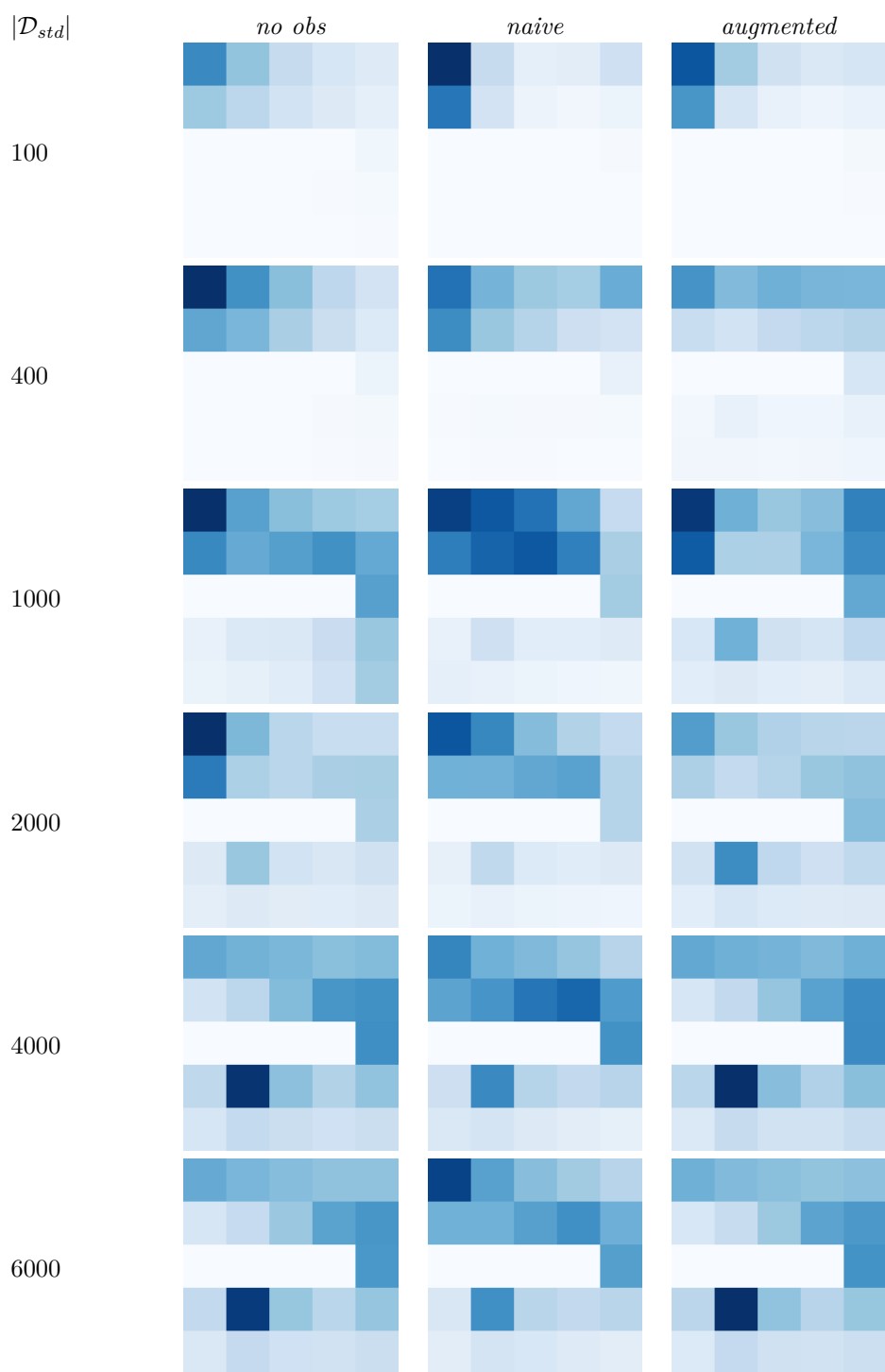

Figure 9: Evolution of the test-time agent trajectories in the `sloppy dark room` experiment. We report a heatmap of the tiles visited by each agent (*no obs*, *naive*, *augmented*) at different time steps (number of interventional samples collected), averaged over 10 RL runs (10 seeds). Eventually all methods manage to consistently overcome the obstacle and reach the target tile. Our *augmented* method converges to this behaviour earlier on during training.

## D   Proof of Theorem 1.

**Theorem 1.** *Assuming $|\mathcal{D}_{prv}| \to \infty$, for any $\mathcal{D}_{std}$ the recovered causal model is bounded as follows:*

$$\prod_{t=0}^{T-1} \hat{q}(o_{t+1}|o_{0\to t}, do(a_{0\to t})) \geq \prod_{t=0}^{T-1} p(a_t|h_t, i=0)p(o_{t+1}|h_t, a_t, i=0), \ and$$

$$\prod_{t=0}^{T-1} \hat{q}(o_{t+1}|o_{0\to t}, do(a_{0\to t})) \leq \prod_{t=0}^{T-1} p(a_t|h_t, i=0)p(o_{t+1}|h_t, a_t, i=0) + 1 - \prod_{t=0}^{T-1} p(a_t|h_t, i=0),$$

$\forall h_{T-1}, a_{T-1}, T \geq 1$ *where* $p(h_{T-1}, a_{T-1}, i=0) > 0$.

*Proof of Theorem 1.* Consider $q(\tau, i) \in \mathcal{Q}$ any distribution that follows our augmented POMDP constraints. As an intermediary step, we will start by proving the following

$$\prod_{t=0}^{T-1} q(o_{t+1}|h_t, a_t, i=1) = \sum_{z_{0\to T}}^{\mathcal{Z}^{T+1}} q(z_0|h_0, i=0) \prod_{t=0}^{T-1} q(z_{t+1}, o_{t+1}|z_t, a_t, h_t, i=0). \tag{6}$$

First, for any $0 \leq t \leq T-1$, we can write the following factorization

$$q(z_t, z_{t+1}, o_{t+1}|h_t, a_t, i=1) = q(z_t|h_t, a_t, i=1)q(z_{t+1}, o_{t+1}|z_t, h_t, a_t, i=1).$$

Because of the augmented POMDP constraints, the independences $Z_t \perp\!\!\!\perp A_t \mid H_t, I=1$ and $Z_{t+1}, O_{t+1} \perp\!\!\!\perp I \mid Z_t, A_t, H_t$ hold in $q$, which further allows us to write

$$q(z_t, z_{t+1}, o_{t+1}|h_t, a_t, i=1) = q(z_t|h_t, i=1)q(z_{t+1}, o_{t+1}|z_t, h_t, a_t, i=0). \tag{7}$$

Then, we directly get

$$q(o_{t+1}|h_t, a_t, i=1) = \sum_{z_t, z_{t+1}}^{\mathcal{Z} \times \mathcal{Z}} q(z_t|h_t, i=1)q(z_{t+1}, o_{t+1}|z_t, h_t, a_t, i=0). \tag{8}$$

Now, let us consider the special case where $T=1$. We can use the constraint $Z_0 \perp\!\!\!\perp I \mid H_0$ to write

$$q(o_1|h_0, a_0, i=1) = \sum_{z_{0\to1}}^{\mathcal{Z}^2} q(z_0|h_0, i=0)q(z_1, o_1|z_0, h_0, a_0, i=0),$$

which is equation (6), the desired result, for $T=1$. In the case where $T \geq 2$, we can reuse equation (8) to write

$$q(o_T|h_{T-1}, a_{T-1}, i=1) = \sum_{z_{T-1\to T}}^{\mathcal{Z}^2} q(z_{T-1}|h_{T-2}, a_{T-2}, o_{T-1}, i=1)q(z_T, o_T|z_{T-1}, h_{T-1}, a_{T-1}, i=0)$$

$$= \sum_{z_{T-1\to T}}^{\mathcal{Z}^2} \frac{q(z_{T-1}, o_{T-1}|h_{T-2}, a_{T-2}, i=1)}{q(o_{T-1}|h_{T-2}, a_{T-2}, i=1)} q(z_T, o_T|z_{T-1}, h_{T-1}, a_{T-1}, i=0)$$

$$\prod_{t=T-2}^{T-1} q(o_{t+1}|h_t, a_t, i=1) = \sum_{z_{T-1\to T}}^{\mathcal{Z}^2} q(z_{T-1}, o_{T-1}|h_{T-2}, a_{T-2}, i=1)q(z_T, o_T|z_{T-1}, h_{T-1}, a_{T-1}, i=0).$$

Then, we can introduce variable $Z_{T-2}$ and use equation (7) again to obtain

$$\prod_{t=T-2}^{T-1} q(o_{t+1}|h_t, a_t, i=1) = \sum_{z_{T-2\to T}}^{\mathcal{Z}^3} q(z_{T-2}, z_{T-1}, o_{T-1}|h_{T-2}, a_{T-2}, i=1)q(z_T, o_T|z_{T-1}, h_{T-1}, a_{T-1}, i=0)$$

$$= \sum_{z_{T-2\to T}}^{\mathcal{Z}^3} q(z_{T-2}|h_{T-2}, i=1) \prod_{t=T-2}^{T-1} q(z_{t+1}, o_{t+1}|z_t, h_t, a_t, i=0).$$

In the case where $T = 2$, we can use $Z_0 \perp\!\!\!\perp I \mid H_0$ again to obtain equation (6), the desired result for $T = 2$. In the case where $T \geq 3$, we can apply the same steps again to obtain

$$\prod_{t=T-3}^{T-1} q(o_{t+1}|h_t, a_t, i = 1) = \sum_{z_{T-3 \to T}}^{\mathcal{Z}^4} q(z_{T-3}|h_{T-3}, i = 1) \prod_{t=T-3}^{T-1} q(z_{t+1}, o_{t+1}|z_t, h_t, a_t, i = 0).$$

Now, either $T = 3$ and we can use $Z_0 \perp\!\!\!\perp I \mid H_0$ to obtain equation (6), or $T \geq 4$ and we can continue the decomposition by introducing $Z_{T-4}$. By following this recursive approach we eventually reach $Z_0$ and prove equation (6) for any $T$.

Let us now re-express equation (6) as follows

$$\prod_{t=0}^{T-1} q(o_{t+1}|h_t, a_t, i = 1) = \sum_{z_{0 \to T}}^{\mathcal{Z}^{T+1}} q(z_0|h_0, i = 0) \left( \prod_{t=0}^{T-1} q(z_{t+1}, o_{t+1}|z_t, h_t, a_t, i = 0) \right) \left( \prod_{t=0}^{T-1} q(a_t|z_t, h_t, i = 0) \right)$$
$$+ \sum_{z_{0 \to T}}^{\mathcal{Z}^{T+1}} q(z_0|h_0, i = 0) \left( \prod_{t=0}^{T-1} q(z_{t+1}, o_{t+1}|z_t, h_t, a_t, i = 0) \right) \left( 1 - \prod_{t=0}^{T-1} q(a_t|z_t, h_t, i = 0) \right) \bigg)$$

$$\prod_{t=0}^{T-1} q(o_{t+1}|h_t, a_t, i = 1) = \prod_{t=0}^{T-1} q(a_t|h_t, i = 0) q(o_{t+1}|h_t, a_t, i = 0)$$
$$+ \sum_{z_{0 \to T}}^{\mathcal{Z}^{T+1}} q(z_0|h_0, i = 0) \left( \prod_{t=0}^{T-1} q(z_{t+1}, o_{t+1}|z_t, h_t, a_t, i = 0) \right) \left( 1 - \prod_{t=0}^{T-1} q(a_t|z_t, h_t, i = 0) \right).$$

By assuming probabilities are positive, we can substitute the second term by 0 to obtain our lower bound

$$\prod_{t=0}^{T-1} q(o_{t+1}|h_t, a_t, i = 1) \geq \prod_{t=0}^{T-1} q(a_t|h_t, i = 0) q(o_{t+1}|h_t, a_t, i = 0).$$

Then by assuming probabilities are upper bounded by 1, we can substitute $q(o_{t+1}|z_{t+1}, z_t, h_t, a_t, i = 0)$ by 1 to obtain our upper bound

$$\prod_{t=0}^{T-1} q(o_{t+1}|h_t, a_t, i = 1) \leq \prod_{t=0}^{T-1} q(a_t|h_t, i = 0) q(o_{t+1}|h_t, a_t, i = 0)$$
$$+ \sum_{z_{0 \to T}}^{\mathcal{Z}^{T+1}} q(z_0|h_0, i = 0) \left( \prod_{t=0}^{T-1} q(z_{t+1}|z_t, h_t, a_t, i = 0) \right) \left( 1 - \prod_{t=0}^{T-1} q(a_t|z_t, h_t, i = 0) \right)$$
$$\leq \prod_{t=0}^{T-1} q(a_t|h_t, i = 0) q(o_{t+1}|h_t, a_t, i = 0) + 1 - \prod_{t=0}^{T-1} q(a_t|h_t, i = 0).$$

Finally, with $\hat{q}$ solution of (5) and $|\mathcal{D}_{prv}| \to \infty$ we have that $D_{\mathrm{KL}}(p(\tau|i = 0)\|\hat{q}(\tau|i = 0)) = 0$, and thus $\hat{q}(a_t|h_t, i = 0) = p(a_t|h_t, i = 0)$ and in particular $\hat{q}(o_{t+1}|h_t, a_t, i = 0) = p(o_{t+1}|h_t, a_t, i = 0)$, which allows us to conclude. $\square$

