# OpenReview forum: "Using Confounded Data in Latent Model-Based Reinforcement Learning"
_TMLR — Accepted by TMLR_

### Review · Reviewer_9ush · 2023-04-13

**Summary Of Contributions:**

The paper discusses an offline reinforcement learning setting in which both interventional and observational (confounded) data are available. The focus of the paper is on a model-based setting, and the paper introduces a latent-variable formulation, which allows a learning procedure that can leverage both the observational and interventional data.

In order, the main contributions of the paper are:
1. formulating the latent-variable MLE learning procedure in Eq. (5),
2. describing an inference procedure for obtaining an estimator (paragraph Inference).
3. proving that the approach gives (a) an unbiased estimator (Prop. 1), and also has better generalization guarantees compared to using interventional data only,
4. demonstrating empirically the method, on a range of examples.

Overall, the paper is well-written, and in most places clear and easy to follow. The key theoretical result is contained in Appendix D, which I was able to check, and I believe the proof is correct (although I make minor suggestions below for how to make the proof easier to follow).

**Audience:**

Yes

**Broader Impact Concerns:**

No major concerns.

**Claims And Evidence:**

Yes

**Requested Changes:**


* adding model-based to the paper title,
* improving the introduction to account for previous work in the field,
* adding more annotation in Appendix D; for instance, the transition from the first display to the second display on page 23 could have a reference to the first display of the proof on page 22; Furthermore, I wonder if the result in the third display on page 23 could be done sequentially? Also, perhaps adding underbrace annotation would help streamline the proof reading, e.g. using $\underbrace{q(\dots)}_{\leq 1}$. I note that this is an interesting proof, and I would not be surprised if other readers will try to go through it. Making such adaptations could help the readers a bit.

**Strengths And Weaknesses:**

## Strengths

The main strengths of the paper are the novel formulation of how to use interventional and observational data in a model-based setting, together with proving generalization guarantees. This clearly makes a step in an interesting direction, and I believe is of interest to the general audience in RL / causal inference. Also, I find the generalization of Manski's bound as interesting. Overall, I am positive about the paper.

## Weaknesses

The largest concern is the framing of the paper, which can be easily amended. In particular, note that:

* the paper title is too broad -- I believe it should contain the words "model-based"
* the abstract and introduction do not mention that the setting is model-based (until the contribution bullets); also, the introduction is clearly missing references to existing works in causality & RL (these references are only discussed at the very end of the paper). From reading the intro, the reader may think this is the very first paper on causality & RL, which is not the case.

Otherwise, I am also wondering if Section 3 is a major contribution of the paper (although I do appreciate Sections 4, 5 which follow from it), and if it should be listed as such in the contribution bullets.

### Minor comments

Typos:
* "they it might get" page 2,
* $p_{obs}$ appears at start of Section 3, not sure if it is defined at this stage? probably just need to say $p_{prv}$,
* in a couple of places, there seem to be spaces before punctuation; would be nice to remove those.

---

> ### Author Response · Authors · 2023-07-14
> **Response**
>
> We thank reviewer 9ush for their valuable comments, their positive feedback, and for checking the proof in the appendix. Below is our point-by-point answer.
>
> > the paper title is too broad -- I believe it should contain the words "model-based"
>
> We agree, and have change the title of the paper to "Using Confounded Data in Latent Model-Based Reinforcement Learning". Thank you for this suggestion.
>
> > the abstract and introduction do not mention that the setting is model-based (until the contribution bullets); also, the introduction is clearly missing references to existing works in causality \& RL (these references are only discussed at the very end of the paper). From reading the intro, the reader may think this is the very first paper on causality \& RL, which is not the case.
>
> Please note that the abstract did mention model-based RL two times (lines 3 and 5). We have now added references to existing works in the first paragraph of the introduction, and have re-worked the second paragraph to highlight more clearly the focus of the paper on model-based RL. Thank you for these suggestion which improve the quality of the paper.
>
> > Otherwise, I am also wondering if Section 3 is a major contribution of the paper (although I do appreciate Sections 4, 5 which follow from it), and if it should be listed as such in the contribution bullets.
>
> We understand and appreciate your concern, however here we would like to kindly disagree. We strongly believe that both the fields of RL and causality share a lot of concepts in common (we argue in the paper that model-based RL is causal inference), and they would benefit a lot from a closer integration. In the paper we purposefully decided to dedicate a substantial amount of space to introduce and relate, formally and intuitively, the frameworks of POMDPs, model-based RL and $do$-calculus. While no new or unexpected result is derived in Section 3, we believe it constitutes a contribution of the paper, as it will help both newcomers and experts in the RL community to make connections with the causality literature, and likewise it can help experts from the causality community relate their works to the model-based RL frameworks and its challenges. Hence, we do believe that Section~3 is a substantial contribution of the paper.
>
>
> > Typos: ...
>
> These have been corrected, thank you for telling us.
>
> > adding model-based to the paper title,
>
> We have renamed the paper "Using Confounded Data in Latent Model-Based Reinforcement Learning".
>
> > improving the introduction to account for previous work in the field,
>
> We now briefly mention previous works in the first paragraph of the introduction, and point readers directly to section 6 for a discussion of related works.
>
> > adding more annotation in Appendix D; for instance, the transition from the first display to the second display on page 23 could have a reference to the first display of the proof on page 22; Furthermore, I wonder if the result in the third display on page 23 could be done sequentially? Also, perhaps adding underbrace annotation would help streamline the proof reading, e.g. using  $\underbrace{q(\dots)}_{\leq 1}$. I note that this is an interesting proof, and I would not be surprised if other readers will try to go through it. Making such adaptations could help the readers a bit.
>
> We have considerably reworked and simplified the proof of theorem 1 in appendix D to make it easier to follow and understand.

---

### Review · Reviewer_iFKF · 2023-04-17

**Summary Of Contributions:**

This paper aims to utilize offline confounded observational data to enhance the sample efficiency of online reinforcement learning (RL). To this end, (i) the authors model the model-based RL with confounded observational data as a causal inference problem and import do-calculus as the underlying method for such inference. (ii) The authors then propose a generic method to learn the transition model based on both offline confounded and online interventional data.

To support their proposed method, (i) the authors first prove that their proposed transition is unbiased and generalizes better than inference with online data (asymptotically). (ii) The authors conducted experiments on a toy example environment and show that their proposed method has better performance.

**Audience:**

Yes

**Broader Impact Concerns:**

N.A.

**Claims And Evidence:**

Yes

**Requested Changes:**

Adding some discussions to address my concerns raised in the above sections can make this paper stronger. I am particularly interested in the identifiability and non-asymptotic behavior of the proposed algorithm (See Question 2 and 3 of the above section).

Minor: It would be better if the authors could discuss more regarding the details of the algorithm (e.g., adding a pseudocode algorithm in the paper). This could help potential audiences adopt the proposed methods to real-world POMDP challenges.

**Strengths And Weaknesses:**

Strength:

1. Utilizing confounded observational data for RL is important and interesting. On the one hand, the authors highlighted that naive utilizing the confounded observational data is not effective, which is also demonstrated in the toy examples. On the other hand, the authors proposed a correct way of utilizing confounded observation and justified its effectiveness via theory and experimentation.

Weakness & Questions:
1. The benefit demonstrated by theory seems limited to me. In particular, the theory shows that the proposed method with privileged observation generalizes better than the proposed method without privileged observation, thus demonstrating the benefit of incorporating confounded observation.  But the latent model is not always required to fit transitions of POMDPs. For instance, it is possible to directly fit the observational model without any latent model [e.g., 1 and plenty of references therein]. Or, one could simply regress observations with respect to history. Thus, the advantage of utilizing confounded data demonstrated by the theory might be limited.
2. The asymptotic theory requires near-infinite confounded observation to establish benefit over methods without confounded observation. Would the cofounded data be useful if their access is also limited?
3. Is the latent transition model identifiable with the observational data for generic POMDPs? As a sanity check, I think the latent state transition is not identifiable based purely on the confounded data. Otherwise, one can recover the counterfactual (the do-probability of observations o_{t+1} given do(a_t) and o_t) by marginalizing the transition over the hidden states. The latent states are also not identifiable based purely on the online interventional data for generic POMDPs. Does their combination allows such identification of latent model and how does that happen?





[1] Liu et al., When Is Partially Observable Reinforcement Learning Not Scary? 2022

---

> ### Author Response · Authors · 2023-07-14
> **Response 1/2**
>
> We thank reviewer iFKF for their valuable comments, to which we reply below.
>
> > The benefit demonstrated by theory seems limited to me. In particular, the theory shows that the proposed method with privileged observation generalizes better than the proposed method without privileged observation, thus demonstrating the benefit of incorporating confounded observation. But the latent model is not always required to fit transitions of POMDPs. For instance, it is possible to directly fit the observational model without any latent model [e.g., 1 and plenty of references therein]. Or, one could simply regress observations with respect to history. Thus, the advantage of utilizing confounded data demonstrated by the theory might be limited.
>
> It is true that our theoretical results only apply to our proposed method, and thus only concerns latent model-based RL. We note that a similar a concern is raised by reviewer 9ush. We believe that a similar approach can be developed without the need for a latent-based transition model, which is straightforward in the case where $T=1$. To address this concern, we propose to rename the manuscript "Using Confounded Data in Latent Model-Based Reinforcement Learning", and to mention the possibility of extending our method to latent-free model-based RL in the Discussions.
>
> > The asymptotic theory requires near-infinite confounded observation to establish benefit over methods without confounded observation. Would the cofounded data be useful if their access is also limited?
>
> Our theoretical results are indeed only asymptotic, and do provide any guarantee in the finite-sample regime, although our experiments do suggest that the finite-sample regime does bring improvements in practice. This concern is shared with reviewer ccoi. We now discuss this limitation in a new "Limitations of the provided guarantees" section (4.5).
>
> > Is the latent transition model identifiable with the observational data for generic POMDPs? As a sanity check, I think the latent state transition is not identifiable based purely on the confounded data. Otherwise, one can recover the counterfactual (the do-probability of observations $o_{t+1}$ given $do(a_t)$ and $o_t$) by marginalizing the transition over the hidden states. The latent states are also not identifiable based purely on the online interventional data for generic POMDPs. Does their combination allows such identification of latent model and how does that happen?
>
> You are right, in general neither the latent space, latent state, or latent transition model $p(s_{t+1} | s_t, a_t)$ are identifiable in POMDPs, be it from observational data, interventional data, or any of their combination. Note that we do not make this claim in the paper. For example, permutations or linear transformations of the latent space are always possible, without affecting the marginalized joint POMDP distribution. However, combinations of data from different regimes A and B, be it interventional + observational, or observational + observational with different control mechanisms, do impose joint constraints on the latent transition model $p(s_{t+1} | s_t, a_t)$ and the observation model $p(o_t|s_t)$. These constraints can in turn restrict the marginalized joint distribution of observations and actions, such as the interventional distribution $p(o_{t_1} | h_t, do(a_t))$ as we show in the paper. As a side note, in some very specific settings these constraints can be so strong that the counterfactuals can also be identified, but in general that's not the case. For example, take our guiding door example, and consider two non-blind experts, where the first expert always press the button that leaves the door closed, and the second expert always press the button that opens the door. By combining these two specific observational regimes, the constraints on the latent transition model (which is still not identifiable) are so strong that the interventional distribution $p(\textit{door} | do(\textit{button}))$ becomes identifiable, as well as the counterfactual distribution $p(\overline{\textit{door}} | do(\overline{\textit{button}}), \textit{door}, do(\textit{button}))$ (here we denote $\textit{door}, do(\textit{button})$ the actual realizations, and $\overline{\textit{door}}, do(\overline{\textit{button}})$ realizations in the imagined, counterfactual world). But, again, in general combinations of observational and interventional distributions do not allow for the identification of the latent space, states, or transition model, nor any counterfactual distribution.

---

> ### Author Response · Authors · 2023-07-14
> **Response 2/2**
>
> > Adding some discussions to address my concerns raised in the above sections can make this paper stronger. I am particularly interested in the identifiability and non-asymptotic behavior of the proposed algorithm (See Question 2 and 3 of the above section).
>
> We now discuss the non-asymptotic behavior of our method in a new "Limitations of the provided guarantees" section (4.5). Regarding identifiability of the latent space and transition model, we do not make any such claim in the paper, therefore we feel that a lengthy discussion would be a little bit out-of-scope for the manuscript. We have simply added a note at the end of Section~4.3 to clarify that our method does not require latent identifiability.
>
> > Minor: It would be better if the authors could discuss more regarding the details of the algorithm (e.g., adding a pseudocode algorithm in the paper). This could help potential audiences adopt the proposed methods to real-world POMDP challenges.
>
> The pseudo-code of the algorithm is given in our experimental section (Algorithm~1). If reviewer iFKF judges it important we can include more details in it, but we are concerned this would hurt readability. Please note that we did release the source code with the paper (https://anonymous.4open.science/r/confounded-data-in-rl), where practitioners can find all the details of the algorithm.

---

### Review · Reviewer_ccoi · 2023-05-22

**Summary Of Contributions:**

The paper presents a method to leverage offline data that could be confounded along with online data to improve sample-efficiency in model-based RL.

Working with a kind of POMDP where the hidden state can affect the action being taken, the paper develops a way to use both the confounded data and the data collected in an online manner. The key idea is to introduce a latent variable $z_t$ to model the hidden state $s_t$, and get "supervision" for $z_t$ from both the confounded and non-confounded data sources. The paper shows clear experimental improvements over the naive modeling approaches and prove some theoretical evidence toward how the proposed method improves.

**Audience:**

Yes

**Broader Impact Concerns:**

All methods that better estimate causal models come with some risk of mis-use. The paper does not contribute risk beyond that.

**Claims And Evidence:**

Yes

**Requested Changes:**


- The authors should provide more intuition about what the results of theorem 1 say about performance improvement.
  - For example, the upper bound can be vacuous for large $T$ and $a_t \perp h_t$ because the only negative term gets very small and the upper bound is a number close to 1; so the upper bound reads something like "probability of trajectory < 1".
 - Can the authors better explain where and how theorem 1 is useful to think about?

Some proof questions : Can you explain the proof step in the following places
- The equation after the sentence "Then for every $t\geq 1$ we can further write .."
- And the following step about "By recursively decomposing every .. "

Maybe for discussion and not necessary for the paper

- The paper would be much better if you were to quantify what "strictly better generalization" is. The proof only tells me that a smaller set of models also match the constraints in the data generation $D_{prv}$, but this only tells me that upper bounds on generalization are better. "Strictly better generalization" implies that there is a separation in the generalization error for the no privilege case can be lower bounded at a larger value than the generalization error achieved by the case where confounded data is available.

I do understand that such theory can get complicated, especially with multi-step objectives. It might be useful to point out the gap in the discussion and cite some causal inference papers that show more direct guarantees and discuss similar ideas : 1. https://arxiv.org/abs/2103.16689, 2. https://arxiv.org/pdf/2011.08047.pdf

Minor

- In equation 5, why are the log probabilities not standardized by the size of the data?
- Not sure "POMDP constraints" are defined well? I inferred it as graph independence constraints but be more specific please.

**Strengths And Weaknesses:**


Strengths
- The experiments show a clear improvement with <200 samples in the non-confounded dataset, demonstrating usefulness in the small-data regime.
- the main paper is well written, and the ideas where clearly articulated
- the method is simple yet empirically effective

My concerns are mainly about the theory being a little loose.

Weaknesses
- Unclear whether "unbiased" is the right word in proposition 1. Did the authors mean consistent in the sense that given infinite data in both $D_{prv}$ and $D_{std}$, $\hat{q}$ will converge the truth?
- The theory statements do not show a quantitative improvement that depends on the size of the $D_{prv}$. Without such statements it is unclear which parts of the algorithm require how much confounded data to improve over the naive/no-privilege case.
- The proofs were hard for me to follow (see questions). More step by step writing out would save readers a lot of time.

---

> ### Author Response · Authors · 2023-07-14
> **Response 2/2**
>
> > In equation 5, why are the log probabilities not standardized by the size of the data?
>
> This is an arbitrary choice, which makes the equation lighter. Solving equation (5) with our without normalization yields the same solution.
>
> > Not sure "POMDP constraints" are defined well? I inferred it as graph independence constraints but be more specific please.
>
> After a quick search, in the manuscript we always say "augmented POMDP constraints". What we mean is all the conditional independence constraints that can be extracted from the augmented POMDP graph via $d$-separation, but also the additional "augmented" contextual conditional independence constraint $A_t \perp S_t \mid H_t,I=1$. We now clarify this concept the first time we mention it, in the proof of Proposition 1.

---

> ### Author Response · Authors · 2023-07-14
> **Response 1/2**
>
> We thank reviewer ccoi for their constructive review of the paper under a formal lens, which will help us improve its quality. Below is a point-by-point response to their comments.
>
> > Unclear whether "unbiased" is the right word in proposition 1. Did the authors mean consistent in the sense that given infinite data in both $\mathcal{D}_\textit{prv}$ and $\mathcal{D}_\textit{std}$, $\hat{q}$ will converge the truth?
>
> Indeed, after reading about consistency vs unbiasedness it seems we meant "consistent estimator" in proposition 1. We have made changes accordingly (in blue in the revised manuscript). Thank you for this suggestion.
>
> > The theory statements do not show a quantitative improvement that depends on the size of the $\mathcal{D}_\textit{prv}$. Without such statements it is unclear which parts of the algorithm require how much confounded data to improve over the naive/no-privilege case.
>
> It is true that our theory statements do not provide any finite-sample guarantee, but only provide intuition of what happens in the asymptotic regime $|\mathcal{D}_\textit{prv}| \to \infty$ (reduced hypothesis space for the estimator, and consistency as $|\mathcal{D}_\textit{std}| \to \infty$). We agree that there would be a lot of value in deriving proper generalization bounds with and without the use of $\mathcal{D}_\textit{prv}$, but as you mentioned the theory can get complicated. We propose to acknowledge this limitation of the paper in a new "Limitations of the provided guarantees" section (4.5), and to leave such derivation as future work.
>
> > The proofs were hard for me to follow (see questions). More step by step writing out would save readers a lot of time.
>
> Thank you for this comment, we have considerably refactored and simplified the proof of Theorem 1, with more step-by-step guidance so that it is easier to follow.
>
> > The authors should provide more intuition about what the results of theorem 1 say about performance improvement. For example, the upper bound can be vacuous for large $T$ and $a_t \perp h_t$ because the only negative term gets very small and the upper bound is a number close to 1; so the upper bound reads something like "probability of trajectory $<$ 1".
>
> We agree, the upper bound in Theorem 1 is rather loose. We know for a fact that it is not the best bound one can obtain, and in the updated manuscript we propose a simple way to expand it into a tighter upper bound that does not vanish to 1 as $T \to \infty$. Still, even this new bound is most likely not tight either. The purpose of Theorem 1 in the paper is merely to serve as a building block in the argument "observational data creates bounds (in the asymptotic regime), therefore the hypothesis space for learning the model is reduced, therefore learning is more efficient". Providing tighter bounds would be nice and would arguably give valuable insights into why the method works, but this would not have much impact on the narrative of the paper, as we don't use these bounds quantitatively in our theoretical argument, in the design of our method, or in our experiments. We propose to acknowledge and discuss this limitation in a new "Limitations of the provided guarantees" section (4.5).
>
> > Can the authors better explain where and how theorem 1 is useful to think about?
>
> See our answer above, and the new "Limitations of the provided guarantees" section (4.5).
>
> > The paper would be much better if you were to quantify what "strictly better generalization" is. The proof only tells me that a smaller set of models also match the constraints in the data generation $\mathcal{D}_\textit{prv}$, but this only tells me that upper bounds on generalization are better. "Strictly better generalization" implies that there is a separation in the generalization error for the no privilege case can be lower bounded at a larger value than the generalization error achieved by the case where confounded data is available.
>
> Thank you for raising this point. What we meant was "strictly better generalization bounds", in the sense that the resulting hypothesis space is a strict subset of the original one, hence any generalization bound must be strictly smaller. We realize that both the wording and also the reasoning were a bit loose, and in particular a strictly smaller hypothesis space does not necessarily yield in a strictly smaller complexity bound. For small enough dataset sizes (e.g., $|\mathcal{D}_\textit{prv}|=1$), the effective hypothesis space does not necessarily change, hence some generalization bounds would not change. We have removed the adverb "strictly" and updated Corollary 1 and its proof accordingly.

---

### Author Response · Authors · 2023-07-14
**General response to reviews**

We are grateful to reviewers ccoi, iFKF and 9ush for their valuable time and insights. We have taken into account their comments and requested changes into the updated revision of the manuscript (changes are highlighted in blue). The major changes are listed below:
 - **Limitations of the theoretical results** (ccoi and iFKF): both reviewers have pointed to limitations in our theoretical results, mostly in the fact that the upper bound in Theorem 1 is not tight and collapses to 1 as $T \to \infty$, and also that our results are only asymptotic and do not provide any guarantee in the finite-sample regime. We have added a new Section~4.5 "Limitations of the provided guarantees" which acknowledges and discusses these points.
 - **Proof of Theorem 1 is hard to read** (ccoi and 9ush): we have considerably reworked and simplified the proof to make it easier to follow.
 - **The paper is restricted to latent model-based RL** (iFKF and 9ush): we have renamed the paper ``Using Confounded Data in Latent Model-Based Reinforcement Learning'', and we have re-written part of the introduction to make it clearer that the scope of the paper is restricted to model-based RL. We also mention now the possibility of extending our method to latent-free model-based RL in the Discussions section.
 - **No mention of previous works in the introduction** (reviewer 9ush): we now briefly mention previous works in the first paragraph of the introduction, and point readers directly to section 6 for a discussion of related works.

We provide a point-by-point answer to each reviewer in individual replies to their review.

---

### Decision · Action_Editors · 2023-07-30

**Recommendation:** Accept with minor revision

**Comment:**

This paper is overall well written and the reviewers generally agree that it is a useful contribution. I concur, but have some reservations about the quality of the theory. The decision is Accept with Minor Revisions, related to the theory, that I will check.

The theory has some rigor issues that should be addressed.
1. Proposition 1 about consistency does not clarify what conditions are required on the probabilistic models. The work [32] is cited about maximum likelihood estimators, to state that MLE will be consistent. But MLE is not always consistent (e.g., what if you lack identifiability?). You should check the conditions for your MLE problem.

2. Corollary 1 needs to be modified. The claim is that you have a strict subset. This is intuitively true, but its not clear it is actually shown. Is it possible for those two sets to be the same? (The one with q(tau|i = 0) = p(tau | i = 0) and without). It is not clear to me why showing that the qhat(stuff) > 0 automatically ensures you have a strict subset.

3. One other minor comment. You assume infinite observational data. It is possible a finite sample result could be given, if you use a high probability result instead. This may not be too difficult to obtain, since you just use D_prv being infinite in the last part of the proof of Theorem 1, to say two distributions are the same. You might instead have a probabilistic claim, and add a small error term to account for a lack of perfect equivalence.

Another option is to avoid writing a formal result in the first place, since that is not the main contribution of this work anyway. It is intuitive that adding the observational data acts like a regularizer. It is also intuitive that MLE should behave as expected. You might simply want to discuss, more informally, why this approach should have the desired properties, without writing explicit propositions and corollaries. You could still keep Theorem 1, if you think it is an important result, but explain a bit more in words why it is important, and maybe even how it relates to the result from Manski (did they also assume infinite observational data? if not, what did they do?).

**Audience:**

Yes.

**Claims And Evidence:**

Mostly yes, with some issues outlined in the theory discussed below.